# Recursive Disentanglement Network

**Yixuan Chen,**[*] **Yubin Shi,**[*] **Dongsheng Li**[†]
{yixuanchen20, ybshi21}@fudan.edu.cn, dongsli@microsoft.com

**Yujiang Wang,**[‡] **Mingzhi Dong**[**]
yujiang.wang14@imperial.ac.uk, mingzhidong@gmail.com

**Yingying Zhao,**[*] **Robert Dick**[§]
yingyingzhao@fudan.edu.cn, dickrp@umich.edu

**Qin Lv,**[¶] **Fan Yang,**[‖] **Li Shang**[*]
qin.lv@colorado.edu, {yangfan, lishang}@fudan.edu.cn

## Abstract

Disentangled feature representation is essential for data-efficient learning. The feature space of deep models is inherently compositional. Existing $\beta$-VAE-based methods, which only apply disentanglement regularization to the resulting embedding space of deep models, cannot effectively regularize such compositional feature space, resulting in unsatisfactory disentangled results. In this paper, we formulate the compositional disentanglement learning problem from an information-theoretic perspective and propose a recursive disentanglement network (RecurD) that propagates regulatory inductive bias recursively across the compositional feature space during disentangled representation learning. Experimental studies demonstrate that RecurD outperforms $\beta$-VAE and several of its state-of-the-art variants on disentangled representation learning and enables more data-efficient downstream machine learning tasks.

## 1 Introduction

Recent progress in machine learning demonstrates the ability to learn disentangled representations is essential for data-efficient learning, such as controllable image generation, image manipulation, and domain adaptation (Suter et al., 2019; Zhu et al., 2018; Peng et al., 2019; Gabbay & Hoshen, 2021; 2019). $\beta$-VAE (Higgins et al., 2017) and its variants are the most investigated approaches for disentangled representation learning. Recent works on $\beta$-VAE-based methods introduce various inductive biases as regularization terms and directly apply them on the resulting embedding space of deep models, such as the bottleneck capacity constraint (Higgins et al., 2017; Burgess et al., 2018), total correlation among variables (Kim & Mnih, 2018; Chen et al., 2018), and the mismatch between aggregated posterior and prior (Kumar et al., 2017), aiming to balance among representation capacity, independence constraints, and reconstruction accuracy. Indeed, as demonstrated by Locatello et al. (2020; 2019), unsupervised disentanglement is fundamentally impossible without explicit inductive biases on models and data sets.

However, our study shows that existing $\beta$-VAE-based methods may not be able to learn satisfactory disentangled representations even for fairly trivial cases. This is due to the fact that the feature spaces of deep models have inherently compositional structures, i.e., each complex feature is a composition of primitive features. However, existing methods with regularization terms solely applied to the resulting embedding space cannot effectively propagate disentangled regularization across such compositional feature space. As shown in Figure 1, applying the standard $\beta$-VAE to the widely

[*] China and Shanghai Key Laboratory of Data Science, School of Computer Science, Fudan University, Shanghai, China, [†] Microsoft Research Asia, Shanghai, China, [‡] Department of Computing, Imperial College London, London, United Kingdom, [§] Department of Electrical Engineering and Computer Science, University of Michigan, Michigan, United States, [¶] Department of Computer Science, University of Colorado Boulder, Boulder, United States, [‖] School of Microelectronics, Fudan University, Shanghai, China, [**] The corresponding author.

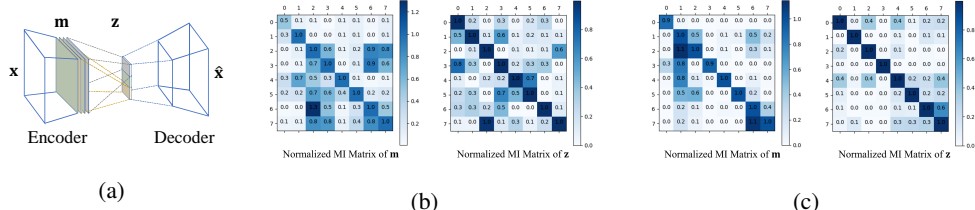

(a)  (b)  (c)

Figure 1: Illustration of the negative impact on ignoring the compositional structure of the representation space using dSprites dataset: (a) Illustration of $\mathbf{z}$ (from embedding space) and $\mathbf{m}$ (from intermediate layers). (b) $\mathbf{z}$ is not disentangled sufficiently when $\mathbf{m}$ is not disentangled sufficiently. (c) The disentanglement of $\mathbf{z}$ improves, as that of $\mathbf{m}$ improves.

used dataset dSprites (Matthey et al., 2017), we visualize the resulting representation $\mathbf{z}$, as well as its compositional low-level representations $\mathbf{m}$ extracted from the previous layer (as shown in Figure 1(a)), and evaluate the independence between each pair of $\mathbf{m}$ and each pair of $\mathbf{z}$, respectively [†]. Figure 1(b) and Figure 1(c) show that the disentanglement quality of low-level features $\mathbf{m}$ may impact the resulting representation $\mathbf{z}$ in terms of disentanglement quality. This study demonstrates the potential benefit to regularize the compositional feature space of deep models during disentangled representation learning.

*This work aims to tackle the compositional disentanglement learning problem.* First, we formulate disentangled representation learning from an information-theoretic perspective, and introduce a new learning objective covering three essential properties for learning disentangled representations: sufficiency, minimal sufficiency, and disentanglement. Theoretical analysis shows that the proposed learning objective is a general form of $\beta$-VAE and several of its state-of-the-art variants. Next, we extend the proposed learning objective to cover the disentanglement representation learning problem in the compositional feature space. Governed by the proposed learning objective, we present Recursive Disentanglement Network (RecurD), a compositional disentanglement learning method, which directs the disentanglement learning process across the compositional feature space by applying regulatory inductive bias recursively through the feed-forward network. We argue that the recursive propagation of inductive bias through the feed-forward network imposes a sufficient condition of disentangled representation learning. Empirical studies demonstrate that RecurD outperforms $\beta$-VAE (Higgins et al., 2017) and several other variants of VAE (Burgess et al., 2018; Kim & Mnih, 2018; Chen et al., 2018; Kumar et al., 2017) on disentangled representation learning and achieves more data-efficient learning in downstream machine learning tasks.

## 2  COMPOSITIONAL DISENTANGLEMENT LEARNING

In this section, we first formulate disentanglement learning from the information-theoretic perspective by introducing three key properties and show such formulation is a general form of the optimization objectives of $\beta$-VAE and several of its variants. Next, we extend the principled objective to the compositional feature space to tackle the compositional disentanglement learning problem.

### 2.1  DISENTANGLEMENT LEARNING OBJECTIVE

The challenge of representation learning can be formulated as finding a distribution $p(\mathbf{z}|\mathbf{x})$ that maps original data $\mathbf{x} \in \mathcal{X}$ into a representation $\mathbf{z}$ with fixed amount of variables $\mathbf{z} = \{\mathbf{z}_1, \ldots, \mathbf{z}_n\}$ (Bengio et al., 2013). The key intuition of $\mathbf{z}$ is to capture minimal sufficient information in a disentangled manner, given the reconstruction task $\mathbf{x} \approx \hat{\mathbf{x}}$. We denote the representation learning process as a Markov Chain for which $\hat{\mathbf{x}} \to \mathbf{x} \to \mathbf{z}$, which means $\mathbf{z}$ depends on $\hat{\mathbf{x}}$ only through $\mathbf{x}$, i.e., $p(\mathbf{z}|\mathbf{x}) = p(\mathbf{z}|\mathbf{x}, \hat{\mathbf{x}})$ (see also, (Cover, 1999; Achille & Soatto, 2018)). The principled properties of $\mathbf{z}$ are defined as follows:

**Definition 1.** *Sufficiency: a representation $\mathbf{z}$ of $\mathbf{x}$ for $\hat{\mathbf{x}}$ is sufficient if $I(\mathbf{x}, \hat{\mathbf{x}}) = I(\mathbf{z}, \hat{\mathbf{x}})$.*

---

[†]The independence between two components $\mathbf{c}_i$ and $\mathbf{c}_j$ is measured by the normalized mutual information (Chen et al., 2018). Whenever $NMI(\mathbf{c}_i; \mathbf{c}_j) = I(\mathbf{c}_i; \mathbf{c}_j)/H(\mathbf{c}) = 0$, $\mathbf{c}_i$ and $\mathbf{c}_j$ are independent (disentangled).

For the reconstruction task, $\mathbf{z}$ is sufficient if it can successfully reconstruct $\mathbf{x}$ by $\hat{\mathbf{x}}$. The difference between $I(\mathbf{x};\hat{\mathbf{x}})$ and $I(\mathbf{z};\hat{\mathbf{x}})$ is computed as follows: $I(\mathbf{x};\hat{\mathbf{x}}) - I(\mathbf{z};\hat{\mathbf{x}}) = I(\mathbf{x};\hat{\mathbf{x}}|\mathbf{z}) = H(\hat{\mathbf{x}}|\mathbf{z}) - H(\hat{\mathbf{x}}|\mathbf{x})$. Given the reconstruction task $\mathbf{x} \approx \hat{\mathbf{x}}$, $H(\hat{\mathbf{x}}|\mathbf{x})$ is constant and independent to $\mathbf{z}$, so the sufficient property can be optimized by minimizing $H(\hat{\mathbf{x}}|\mathbf{z})$ (Federici et al., 2020; Dubois et al., 2020).

**Definition 2.** *Minimal Sufficiency: a representation $\mathbf{z}$ of $\mathbf{x}$ is minimal sufficient if $I(\mathbf{x};\mathbf{z}) = I(\mathbf{z};\hat{\mathbf{x}})$.*

A minimal sufficient $\mathbf{z}$ encodes the minimum amount of information about $\mathbf{x}$ required to reconstruct $\hat{\mathbf{x}}$ (Cover, 1999; Achille & Soatto, 2018). Since $I(\mathbf{z};\hat{\mathbf{x}})$ equals to $I(\mathbf{x};\hat{\mathbf{x}})$ when $\mathbf{z}$ is sufficient, the difference is computed as $I(\mathbf{x};\mathbf{z}) - I(\mathbf{z};\hat{\mathbf{x}}) = I(\mathbf{x};\mathbf{z}) - I(\mathbf{x};\hat{\mathbf{x}})$. Given the reconstruction task $\mathbf{x} \approx \hat{\mathbf{x}}$, $I(\mathbf{x};\hat{\mathbf{x}})$ is constant and independent to $\mathbf{z}$, so the minimal sufficiency property can be optimized by minimizing $I(\mathbf{x};\mathbf{z})$.

**Definition 3.** *Disentanglement: a representation denoted as $\mathbf{z} = \{\mathbf{z}_1, \ldots, \mathbf{z}_n\}$ is disentangled if $\sum_{j \neq i} I(\mathbf{z}_i;\mathbf{z}_j) = 0$.*

From the definition of mutual information, $I(\mathbf{z}_i;\mathbf{z}_j) = H(\mathbf{z}_i) - H(\mathbf{z}_i|\mathbf{z}_j)$ denotes the reduction of uncertainty in $\mathbf{z}_i$ when $\mathbf{z}_j$ is observed (Cover, 1999). If any two components $\mathbf{z}_i$ and $\mathbf{z}_j$ are disentangled, changes to $\mathbf{z}_i$ have no influence on $\mathbf{z}_j$, which means $I(\mathbf{z}_i;\mathbf{z}_j) = 0$.

A representation satisfying all these properties can be found by introducing two Lagrange multipliers $\lambda_1$ and $\lambda_2$ for two constrained expected properties with respect to the fundamental sufficiency property. The principled objective of disentanglement learning is to minimize the following objective:

$$\mathcal{L} = H(\hat{\mathbf{x}}|\mathbf{z}) + \lambda_1 I(\mathbf{x};\mathbf{z}) + \lambda_2 \sum_{j \neq i} I(\mathbf{z}_i, \mathbf{z}_j). \tag{1}$$

The above objective can be interpreted as the reconstruction error, plus two regularizers that yield an optimally disentangled representation. The principled objective also helps us analyze and understand the success of recently developed $\beta$-VAE-based methods. These methods operate with an encoder with parameter $\phi$ and a decoder with parameter $\theta$, to induce the joint distributions $q(\mathbf{x},\mathbf{z}) = q_\phi(\mathbf{z}|\mathbf{x})q(\mathbf{x})$ and $p(\mathbf{x},\mathbf{z}) = p_\theta(\mathbf{x}|\mathbf{z})p(\mathbf{z})$, respectively, where $p(\mathbf{z})$ is a fixed prior distribution. The learning objective of $\beta$-VAE contains the reconstruction error and KL divergence between the variational posterior and prior. To understand the relationship of learning objectives between Equation 1 and $\beta$-VAE-based methods, we decompose $I(\mathbf{x};\mathbf{z})$ (Kim & Mnih, 2018) and estimate an upper bound for $\sum_{j \neq i} I(\mathbf{z}_i, \mathbf{z}_j)$ (Te Sun, 1980; 1975), then we assign different weights as follows:

$$\lambda_1 I(\mathbf{x};\mathbf{z}) + \lambda_2 \sum_{j \neq i} I(\mathbf{z}_i, \mathbf{z}_j)$$

$$\leq \lambda_a \mathbf{E}_{\mathbf{x}}\left[KL\left(q(\mathbf{z}|\mathbf{x}) \| p(\mathbf{z})\right)\right] + \lambda_b KL\left(q(\mathbf{z}) \| \prod_{j=1}^n q(\mathbf{z}_j)\right) + \lambda_c \sum_{j=1}^n KL\left(q(\mathbf{z}_j) \| p(\mathbf{z}_j)\right). \tag{2}$$

As shown in Table 1, *the learning objectives of $\beta$-VAE and its four variants can be regarded as specific cases of Equation 1*, i.e., they assign different weights to our regularization terms, which can balance among latent variables capacity, independence constraints, and reconstruction accuracy, leading to successful disentangled representation learning (Zhao et al., 2017; Li et al., 2020). More details can be found in Appendix B. However, in these works, the inductive bias toward disentanglement is only applied to the embedding space of $\mathbf{z}$, ignoring the need of disentanglement during feature composition in feed-forward networks.

Table 1: Deriving the learning objectives of $\beta$-VAE and its four variants as specific cases of Equation 1.

| Method | $\beta$-VAE | FactorVAE | $\beta$-TCVAE | DIP-VAE | InfoVAE |
|--------|-------------|-----------|---------------|---------|---------|
| Weight Relation | $\lambda_a = \beta$. | $\lambda_a = 1$, $\lambda_b = \gamma$. | $\lambda_a = 1$, $\lambda_b = \beta$. | $\lambda_a = 1$, $\lambda_b = \lambda_c = \lambda$. | $\lambda_a = 1 - \alpha$, $\lambda_b + \lambda_c = \alpha + \lambda - 1$. |

## 2.2 COMPOSITIONAL OBJECTIVE

Considering an encoder with $L$ layers to encode original data $\mathbf{x}$ into disentangled representation $\mathbf{z}$. Let us denote $\mathbf{m}^l$ as the input features of the $l$-layer, which are divided into groups of features, i.e., $\mathbf{m}^l = \cup_j \mathbf{m}^l_j$, where $\mathbf{m}^l_j$ is the $j$-th feature subset.

We formulate the compositional relation between features from two consecutive layers as follows: $\mathbf{m}_j^{l+1} = Layer\left(\mathbf{m}^l \times \mathbf{w}_j^l\right)$. $Layer$ can be applicable to commonly used neural network layers, e.g., the convolution layer in computer vision tasks. The compositional relation is achieved by a composition matrix $\mathbf{w}^l \in \mathbb{R}^{d_l \times d_{l+1}}$, so that features in $\mathbf{m}^l$ are divided into $d_{l+1}$ groups after passing through all compositional vectors ($\mathbf{w}_j^l$s). Note that $\mathbf{m}_j^{l+1}$ is only related to the subset of features from $\mathbf{m}^l$ selected by $\mathbf{w}_j^l$.

Similar to Section 2.1, we assume that the learning process of $\mathbf{m}^{l+1}$ depends on $\mathbf{m}^l$ through original data $\mathbf{x}$, denoted as the Markov chain: $\mathbf{m}^l \rightarrow \mathbf{x} \rightarrow \mathbf{m}^{l+1}$. It can be written as $\mathbf{m}^{l+1} \rightarrow \mathbf{x} \rightarrow \mathbf{m}^l$ according to the conditional independence implied by the Markovity (Cover, 1999). Consider two output features $\mathbf{m}_i^{l+1}, \mathbf{m}_j^{l+1} \in \mathbf{m}^{l+1}$ and their corresponding input feature subsets $\mathbf{m}_i^l, \mathbf{m}_j^l \in \mathbf{m}^l$, we define key notions as follows:

**Definition 4.** *Compositional Disentanglement:* $\mathbf{m}_i^l$ *and* $\mathbf{m}_j^l$ *are disentangled if* $I\left(\mathbf{m}_i^l; \mathbf{m}_j^l\right) = 0$.

A disentangled representation of $\mathbf{m}_i^l$ and $\mathbf{m}_j^l$ may improve the disentanglement quality between $\mathbf{m}_i^{l+1}$ and $\mathbf{m}_j^{l+1}$. Similar to Definition 3, we can achieve compositional disentanglement by minimizing $I\left(\mathbf{m}_i^l; \mathbf{m}_j^l\right)$.

**Definition 5.** *Compositional Minimal Sufficiency: Assume that the learning process of* $\mathbf{m}_j^{l+1}$ *is denoted by the Markov chain:* $\mathbf{m}_j^{l+1} \rightarrow \mathbf{x} \rightarrow \left(\mathbf{m}_i^l, \mathbf{m}_j^l\right)$. *Given the original data* $\mathbf{x}$, *an input feature set* $\mathbf{m}_j^l$ *for the output feature* $\mathbf{m}_j^{l+1}$ *is minimal sufficient if* $I\left(\mathbf{x}; \mathbf{m}_j^{l+1}\right) = I\left(\mathbf{m}_j^l; \mathbf{m}_j^{l+1}\right)$.

For the output feature $\mathbf{m}_j^{l+1}$, the input feature set $\mathbf{m}_j^l$ is sufficient and another input feature set $\mathbf{m}_i^l$ is superfluous when $\mathbf{m}_j^l$ is able to capture all information of $\mathbf{m}_j^{l+1}$ as well as the original data $\mathbf{x}$. Furthermore, according to Data-Processing Inequality (DPI) (Cover, 1999; Achille & Soatto, 2018) in the Markov chain, there exists an inequality that:

$$I\left(\mathbf{x}; \mathbf{m}_j^{l+1}\right) \geq I\left(\mathbf{m}_j^{l+1}; \mathbf{m}_j^l\right) + I\left(\mathbf{m}_j^{l+1}; \mathbf{m}_i^l\right) - I\left(\mathbf{m}_i^l; \mathbf{m}_j^l\right), \qquad (3)$$

where the difference between $I\left(\mathbf{x}; \mathbf{m}_j^{l+1}\right)$ and $I\left(\mathbf{m}_j^{l+1}; \mathbf{m}_j^l\right)$ is equivalent to the difference between $I\left(\mathbf{m}_j^{l+1}; \mathbf{m}_i^l\right)$ and $I\left(\mathbf{m}_i^l; \mathbf{m}_j^l\right)$. Therefore, matching $I\left(\mathbf{m}_j^{l+1}; \mathbf{m}_i^l\right)$ to $I\left(\mathbf{m}_i^l; \mathbf{m}_j^l\right)$ can yield a minimal sufficient representation $\mathbf{m}_j^l$ for $\mathbf{m}_j^{l+1}$. Based on the definition of compositional disentanglement, we can optimize the minimal sufficiency by forcing $I\left(\mathbf{m}_j^{l+1}; \mathbf{m}_i^l\right)$ to be 0. More details can be found in Appendix B.

To learn disentangled representation via effectively regularizing the compositional feature space, we augment the principled learning objective (Equation 1) with compositional regularizers. Therefore, the compositional learning objective for disentangled representation is defined as follows:

$$\mathcal{L} = \underbrace{H\left(\hat{\mathbf{x}}|\mathbf{m}^{L+1}\right)}_{\text{sufficient}} + \lambda_1 \underbrace{\left(\sum_{l=2}^{L} \sum_{j \neq i}^{d_{l+1}} I\left(\mathbf{m}_i^l; \mathbf{m}_j^{l+1}\right)\right)}_{\text{minimal sufficient}} + \lambda_2 \underbrace{\left(\sum_{l=2}^{L+1} \sum_{j \neq i}^{d_{l+1}} I\left(\mathbf{m}_i^l, \mathbf{m}_j^l\right)\right)}_{\text{disentangled}}, \qquad (4)$$

where $\mathbf{m}^{L+1}$ denotes the final disentangled representation $\mathbf{z}$. Our intuition is that disentangled learning for compositional feature space could benefit the disentanglement learning for high-level representations.

## 3 RECURSIVE DISENTANGLEMENT NETWORK

We now describe a learning method with the goal of optimizing the compositional disentanglement learning objective. This method, called Recursive Disentanglement Network (RecurD), propagates inductive bias (disentanglement) recursively across the compositional feature space.

### 3.1 MODEL ARCHITECTURE

As shown in Figure 2, RecurD contains an encoder and a decoder to learn the disentangled representation $\mathbf{z}$ of data $\mathbf{x}$ and to reconstruct $\hat{\mathbf{x}}$, where the encoder contains multiple *Recursive Modules* and the decoder is a Deconvolutional Neural Network (Zeiler & Fergus, 2014).

Figure 2: Recursive Disentanglement Network

The first Recursive Module of the encoder is implemented by a multi-channel convolutional network (Lawrence et al., 1997) to encode the original image $\mathbf{x}$. Following the notation in Section 2.2, the output of the 1-st Recursive Module is denoted as $\mathbf{m}^2$, which is also the input of 2-nd Recursive Module. As for the $l$-th ($l \geq 2$) Recursive Module, it contains (1) a *Router* $R$ to learn a composition matrix $\mathbf{w}^l$ from the input features $\mathbf{m}^l$ to decompose $\mathbf{m}^l$ into subsets; and (2) a *Group-of-Encoders (GoE)* layer consisting of $n$ encoders to induce the output feature $\mathbf{m}^{l+1}$.

Inspired by the Gate of Mixture-of-Experts (Shazeer et al., 2017; Fedus et al., 2021) on parameter selection for each input, we present a Router with $TopK$ to learn the compositional relation. In detail, the Router takes $\mathbf{m}^l$ as input and compute composition matrix $\mathbf{w}^l$ via learning similarity:

$$\mathbf{w}^l = \mathrm{softmax}\left(TopK\left(R\left(\mathbf{m}^l\right), k\right)\right), R\left(\mathbf{m}^l\right) = \mathrm{softmax}\left(\left[\mathbf{m}^l\right]\left[\mathbf{m}^l\right]^T\right) \circ \mathbf{v}, \text{and}$$

$$TopK\left(R\left(\mathbf{m}^l\right), k\right) = \begin{cases} R\left(\mathbf{m}^l\right)_{ij}, & \text{if } R\left(\mathbf{m}^l\right)_{ij} \text{ is in the top } k \text{ elements of } R\left(\mathbf{m}^l\right)_{\cdot j} \\ -\infty, & \text{otherwise.} \end{cases} \quad (5)$$

Here, $R\left(\mathbf{m}\right)$ denotes the similarity matrix and $\mathbf{v}$ is a learning matrix learned by a linear layer. $\circ$ denotes Hadamard product. $TopK$ is the compositional strategy to determine whether input feature $\mathbf{m}^l_{ij}$ belongs to the input feature set of $\mathbf{m}^{l+1}_j$, and $k$ is a hyperparameter. The GoE layer consists of $d_{l+1}$ parallel encoders $\{Enc_1, \ldots, Enc_{d_{l+1}}\}$ to generate the output features $\mathbf{m}^{l+1}$, where each encoder is implemented by a convolutional neural network with specific parameters:

$$\mathbf{m}^{l+1}_j = Enc_j\left(\mathbf{m}^l \times \mathbf{w}^l_j\right). \quad (6)$$

Then, $\mathbf{m}^{l+1}$ is obtained by concatenating the outputs from each encoder, which is converted directly to the input of the $l + 1$-th Recursive Module. Note that the output of $L$-th Recursive Module is the learned disentangled representation $\mathbf{z}$. Finally, the Decoder $Dec$ takes $\mathbf{z}$ as input to obtain the reconstructed input $\hat{\mathbf{x}}$: $\hat{\mathbf{x}} = Dec\left(\mathbf{z}\right)$.

## 3.2 LEARNING OF RECURD

As mentioned in Equation 4, the learning objective of RecurD governs the recursive disentanglement learning across the compositional feature space. As shown by related works, a precise estimation of mutual information in the high dimensional space is important for accurately estimating the loss. According to the recent progresses, the mutual information between two representations can be maximized by using any sample-based differentiable mutual information lower bound. Similar to the work of Federici et al. (2020); Hjelm et al. (2019); Wen et al. (2020), we utilize MINE estimators (Belghazi et al., 2018) to estimate the mutual information. This approach introduces an auxiliary parametric model Estimator Net which is jointly optimized during the training procedure using re-parametrized samples from the posterior distribution.

## 4 RELATED WORK

$\beta$-VAE-based methods introduce various regularization terms and directly apply them on the resulting embedding space. Higgins et al. (2017) proposed $\beta$-VAE method by introducing a constraint $\beta$ over the $KL$ divergence between the inferred distribution $q_\phi(\mathbf{z}|\mathbf{x})$ and its prior–an isotropic unit Gaussian. Burgess et al. (2018) found that the network specializes in the factor that contributes most to a small reconstruction error, with a limited channel capacity. Thus, they proposed the Annealed-VAE, an extension of $\beta$-VAE, which gradually adds more latent encoding capacity by enforcing the $KL$ divergence to be at a controllable value $C$. Kim & Mnih (2018) analyzed the disentanglement

performance of $\beta$-VAE by breaking down the regularization term into two components. One is Total Correlation (TC) (Watanabe, 1960), which encourages the marginal distribution of representations to be factorial. The other is mutual information $I(\mathbf{x}; \mathbf{z})$, which reduces the amount of information about $\mathbf{x}$ stored in $\mathbf{z}$. Based on the decomposition, they proposed FactorVAE to relax the regularization of $I(\mathbf{x}; \mathbf{z})$ while directly penalizing the TC term. Similarly, Chen et al. (2018) proposed $\beta$-TCVAE by decomposing the TC penalty as the dependence among variables and the distance between each variable's posterior and prior. Since TC is intractable, both FactorVAE and $\beta$-TCVAE use approximation methods. Another TC-related approach is DIP-VAE (Kumar et al., 2017), whose designers argued that the estimation of TC requires additional parameters and suffers from vanishing gradients. Therefore, DIP-VAE optimizes the moments distance between the aggregated posterior and a factorized prior instead of estimating TC. The above methods improved over $\beta$-VAE by applying inductive biases directly to the high-level latent variable space. However, the feature space of deep models is compositional in nature. Existing variants of $\beta$-VAE cannot effectively apply disentanglement regularization across such compositional feature space, and yield inferior disentanglement in representation learning. Our approach diverges from prior works by taking a principled information-theoretic approach to formulate and analyze the compositional disentanglement feature structure.

Recent works on hierarchical VAEs introduce layer-wise disentanglement regularization to learn conditioning structures across multi-layer latent variables, such as VampPrior (Tomczak & Welling, 2018), Ladder VAEs (Sønderby et al., 2016), and NVAE (Vahdat & Kautz, 2020). In these hierarchical model structures, e.g., the cross-layer residual connection structure in NVAE and VampPrior, inter-layer regularization is less of a focus. The latent variables of a preceding layer serve as shared inputs to the next layer, which introduces information redundancy and hence impairs representation disentanglement. In contrast, the proposed compositional objective optimizes the statistical independence of inter-layer and intra-layer latent variables simultaneously, thereby minimizing the information redundancy of inter-layer information sharing and improving disentanglement quality.

## 5 EXPERIMENTS

This section presents both quantitative and qualitative experiments to evaluate RecurD in terms of disentanglement quality and data efficiency on downstream tasks.

### 5.1 PERFORMANCE OF DISENTANGLEMENT LEARNING

We compare the disentanglement learning performance of RecurD with $\beta$-VAE and its state-of-the-art variants, including $\beta$-VAE (Higgins et al., 2017), Annealed-VAE (Burgess et al., 2018), Factor-VAE (Kim & Mnih, 2018), $\beta$-TCVAE (Chen et al., 2018), DIP-VAE (Kumar et al., 2017). More experimental results can be found in Appendix E.

**Datasets**: We consider two datasets in which each image is obtained by a deterministic function of ground-truth factors: *dSprites* (Matthey et al., 2017) and *3DShapes* (Burgess & Kim, 2018). dSprites contains $737, 280$ binary $64 \times 64$ images of 2D shapes with $5$ ground truth factors, i.e., $3$ shapes, $6$ scales, $40$ orientations, $32$ x-positions, $32$ y-position. 3DShapes contains $480, 000$ RGB $64 \times 64 \times 3$ images of 3D shapes with $6$ ground truth factors, i.e., $4$ shapes, $8$ scales, $15$ orientations, $10$ floor hues, $10$ wall hues, $10$ object hues. For all experiments, we use a 9:1 training to testing data ratio, following earlier work (Kumar et al., 2017; Locatello et al., 2019). More details on network architecture and hyperparameter settings are included in Appendix D.

**Evaluation Metrics for Disentanglement**: There is no standard metric for evaluating disentanglement (Zhou et al., 2021; Ridgeway & Mozer, 2018), and most existing metrics involve the estimation of a variable-factor matrix relating the factors of variation to the learned representations. In the experiments, we consider three widely used metrics: Separated Attribute Predictability (SAP) (Kumar et al., 2017), Mutual Information Gap (MIG) (Chen et al., 2018) and Disentanglement, Completeness, and Informativeness (DCI) (Eastwood & Williams, 2018). DCI contains three metrics for disentanglement (DCI-D), Completeness (DCI-C) and Informativeness (DCI-I), respectively. Due to the space limitation, we present the results of DCI-D and include the results on DCI-C and DCI-I in the Appendix E. Overall, these metrics can comprehensively evaluate RecurD from different disentanglement measurements.

### 5.1.1 QUANTITATIVE RESULTS

For quantitative analysis, we conduct three sets of experiments: 1) evaluating the average performance of disentanglement score via three evaluation metrics; 2) measuring the trade-off among reconstruction, minimal sufficiency and disentanglement with mini-batch samples; and 3) analyzing the influence of compositional objective for disentanglement learning via the three properties.

Table 2 compares the reconstruction error and three widely-used disentanglement metrics of RecurD and five other methods on the dSprites and 3DShapes datasets. Compared with all the baselines, RecurD achieves much lower reconstruction error as well as higher SAP, MIG, and DCI scores in most of the cases. It is important to point out that the reconstruction error of $\beta$-VAE increases as $\beta$ increases (stronger disentanglement regularization), which indicates that $\beta$-VAE comprises reconstruction error for disentanglement. However, RecurD achieves better in both reconstruction and disentanglement, demonstrating the superiority of the proposed compositional learning objective and the recursive disentanglement network.

Figure 3 shows scatter plots relating *Minimal Sufficiency* and *Disentanglement* to reconstruction error (*Sufficiency*), where each point represents a mini-batch of data. Here, we compute the minimal sufficiency score using $I(\mathbf{x}; \mathbf{z})$ and present it by the area of each point. In all baseline methods, we observe that smaller scatters are correlated with higher reconstruction errors and higher disentanglement scores. The reason is that $\beta$-VAE and its variants only place disentanglement regularization on the embedding space of $\mathbf{z}$ which imposes a limit on the capacity of the information channel (Higgins et al., 2017), so that features which are important for reconstruction but harmful for disentanglement may lose during training, leading to compromised reconstruction quality. However, RecurD reveals the necessity of disentanglement during feature composition in feed-forward networks and induces effective information compression. Therefore, compared to all the baselines, the representations from RecurD obtain better disentanglement and minimal sufficiency without degrading reconstruction quality.

Figure 4 shows the impact of the number of Recursive Modules in disentanglement learning. We evaluate three RecurD variants on the principled properties, including $RecurD$ 0, $RecurD$ 1 and $RecurD$ 2 with 1, 2 and 3 Recursive Modules, respectively. And we report the values correspond-

Table 2: Three widely used evaluation scores and reconstruction error on the test sets for dSprites and 3Dshapes. Boldface indicates the best results, i.e., reconstruction error or disentanglement scores.

| dataset | dSprites | | | | 3DShapes | | | |
|---|---|---|---|---|---|---|---|---|
| | Reconst. Error | SAP | MIG | DCI-D | Reconst error | SAP | MIG | DCI-D |
| $\beta$-VAE($\beta = 4$) | 0.0066 | 0.0284 | 0.2617 | 0.1191 | 0.0216 | 0.1463 | 0.2519 | 0.4682 |
| $\beta$-VAE($\beta = 16$) | 0.0094 | 0.0242 | 0.2241 | 0.1820 | 0.0544 | 0.1488 | 0.2629 | 0.4528 |
| $\beta$-VAE($\beta = 60$) | 0.0127 | 0.0445 | 0.1432 | 0.1291 | 0.0624 | 0.1041 | 0.2402 | 0.4317 |
| Annealed-VAE | 0.0171 | 0.0311 | 0.1177 | 0.1449 | 0.0811 | 0.0730 | 0.2217 | 0.4279 |
| Factor-VAE | 0.0228 | 0.0436 | 0.2594 | 0.1955 | 0.0800 | 0.1331 | 0.2630 | 0.4491 |
| $\beta$-TCVAE | 0.0162 | 0.0352 | 0.1585 | 0.1774 | 0.0312 | 0.0364 | 0.2070 | 0.4487 |
| DIP-VAE | 0.0213 | 0.0261 | 0.0731 | 0.1038 | 0.0213 | **0.2013** | **0.3108** | 0.4853 |
| **RecurD** | **0.0047** | **0.0502** | **0.2707** | **0.3841** | **0.0083** | 0.1979 | 0.3105 | **0.5804** |

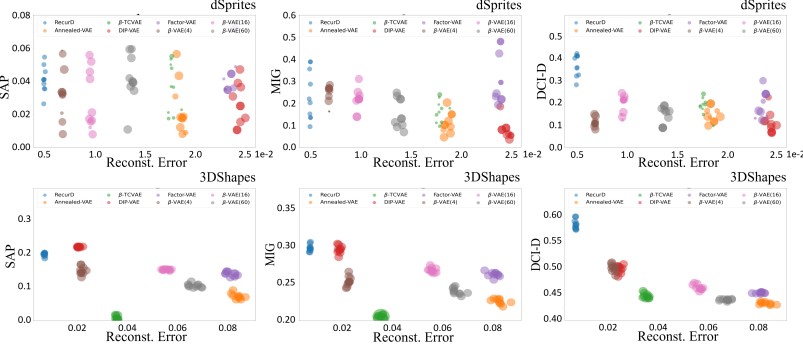

Figure 3: Reconstruction error vs. disentanglement performance. Scatters located at the left top indicate better performance. The area of each scatter represents the minimal sufficiency score estimated by $I(\mathbf{x}; \mathbf{z})$, and smaller area indicates better performance.

ing to the three terms in our compositional learning objective on the training sets. We observe that $RecurD$ 1 and $RecurD$ 2 perform much better than $RecurD$ 0 on the optimization of minimal sufficiency and disentanglement. In addition, during the early stage of training, the performance of $RecurD$ 2 improves much faster than $RecurD$ 1. This experiment demonstrates that the recursive propagation of inductive bias through the feed-forward network improves disentangled representation learning.

### 5.1.2 QUALITATIVE RESULTS

For qualitative analysis, we present the latent traversals of RecurD on two datasets in Figure 5, in which we vary a single variable learned by an encoder in GoE while keeping all others fixed. For images in 3DShapes, the latent traversals show that RecurD is able to successfully capture all the six factors of variation. For images in dSprites, we observe that RecurD is able to discover x-position, y-position and scale (continuous variables). More importantly, RecurD can, to some extent, discover shape and orientation (discrete variables), which have been proved to be struggling for many other methods (Kumar et al., 2017; Kim & Mnih, 2018;

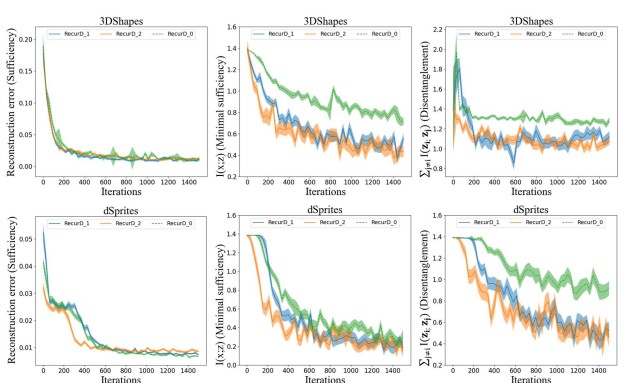

Figure 4: The performance of RecurD with varying number of recursive modules on the principled properties.

Locatello et al., 2019). The reason is that $\beta$-VAE and its variants encourage independence among variables via controlling the information capacity of $\mathbf{z}$ and matching the posterior $p(\mathbf{z}|\mathbf{x})$ to an isotropic unit Gaussian. However, learning discrete variables would require using a discrete prior instead of Gaussian (Kim & Mnih, 2018). On the other hand, RecurD is able to model both discrete and continuous factors by directly disentangling based on the three principled properties on the entire compositional feature space, leading to stronger representation capability.

### 5.1.3 ABLATION AND PARAMETER DEPENDENCE STUDY

In this section, we evaluate the impacts of hyperparameters in both learning objective and model architecture. First, we study the impact of compositional learning objective with varying regularization coefficients ($\lambda_1$ and $\lambda_2$) of minimal sufficiency and disentanglement. Then, we evaluate the influence of hyperparameter $k$ — the group size in the GoE of Recursive Module. Figure 6 (RecurD with varying $\lambda_1$, $\lambda_2$ and $k$) shows the scatter plots of disentanglement score (SAP) along with reconstruction error on the test sets of dSprites and 3DShapes.

As shown in Figure 6, larger penalties on both minimal sufficiency and disentanglement yield higher disentanglement score and lower reconstruction error, demonstrating the importance of the minimal sufficiency term and the disentanglement term in Equation 1. Note that RecurD with $\lambda_2 = 0$ is

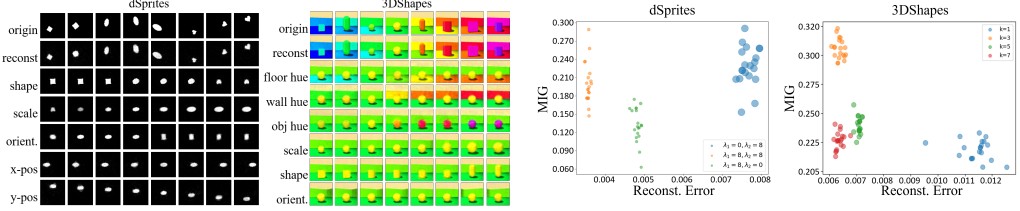

Figure 5: First row: original images. Second row: reconstructions. Remaining rows: reconstructions of latent traversals.

Figure 6: Ablation study on $\lambda_1$, $\lambda_2$ and group size $k$. Note that RecurD with $\lambda_2 = 0$ reduces to $\beta$-VAE with the compositional architecture.

reduced to a standard $\beta$-VAE with the compositional architecture (as $I(\mathbf{x}; \mathbf{z})$ is an lower bound of $KL(p(\mathbf{z}|\mathbf{x})\|p(\mathbf{z})))$, which underperforms RecurD, indicating that an explicit penalty on disentanglement is important for disentangled representation learning. As for the group size in the GoE — $k$, it is not surprising that when $k$ increases, RecurD has lower reconstruction errors, with a slightly inferior disentanglement performance. The reason is that a dense composition of feature space can help the model maintain sufficient information but make it difficult to disentangle latent variables. RecurD with $k = 1$ results in poor performance on both reconstruction and disentanglement, indicating that over-emphasizing decomposition in the feature space may fail to preserve sufficient information. More results are presented in the Appendix E.

## 5.2 PERFORMANCE OF DOWNSTREAM TASKS

In this section, we compare the performance of RecurD and five baseline methods by measuring their data efficiency on two downstream tasks: a standard classification task on the MNIST dataset and a domain generalization task on the MNIST-Rotation dataset (Ghifary et al., 2015). MNIST-Rotation is a synthetic dataset consisting of 6 domains, each containing $1,000$ images of the 10 digits randomly selected from the training set of MNIST, with 6 rotation degrees: $0°$, $15°$, $30°$, $45°$, $60°$ and $75°$. For both tasks, we set the training sets of different proportions: 20%, 40%, 60%, 80%, and 100%, to evaluate the classification performance of RecurD and the baselines with different amounts of training data. For the domain generalization problem, we follow the previous

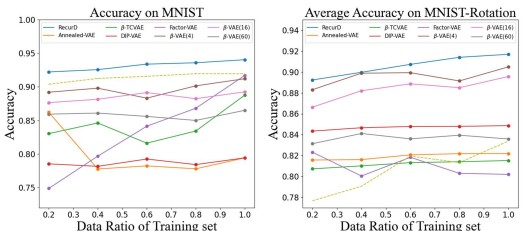

Figure 7: Performance comparison of RecurD and baselines on the standard classification task (left) and the domain generalization task (right). Each model is trained by varying ratios of training data. The dotted line represents the performance of a single-layer neural network.

works (Li et al., 2017; Balaji et al., 2018; Du et al., 2020) with the same train-test split strategy and the leave-one-domain-out strategy, i.e., we take the samples from one domain as the target domain for testing, and the samples from the remaining domains as the source domain for training.

Figure 7 reports the average accuracy of different methods on the same test set. We can observe that all methods achieve decent data efficiency on both tasks, i.e., without much performance degradation even with 20% of training data, suggesting that the learning process of disentangled representation can more effectively capture information from the inputs. However, $\beta$-VAE and its variants do not achieve satisfactory classification accuracy for either task compared to a single-layer neural network. The reason is that $\beta$-VAE and its variants obtain the disentangled representation by limiting the capacity of information channels, thus they tend to only maintain features that contribute more to disentanglement, which may lose informative features that contribute more to classification. Compared to the baselines, RecurD achieves consistently better performance in both tasks, especially on the harder domain generalization task. The reason is that RecurD can learn disentangled representations without sacrificing the reconstruction performance, confirming the hypothesis that compositional disentanglement learning yields better generalization and more data-efficient representations. We believe that the informative disentangled representations emerge when the right balance is achieved between sufficient information preservation and minimal sufficient information learned in a disentangled manner.

## 6 CONCLUSION

This paper has described a solution to the compositional disentangled representation learning problem. We first presented a general information-theoretic formulation of disentanglement representation learning, and then extended it to the compositional feature space. We then described RecurD, a recursive disentanglement network, which propagates regulatory inductive bias recursively across the compositional feature space. RecurD outperforms $\beta$-VAE and its state-of-the-art variants on disentangled representation learning and achieves more data-efficient learning in downstream tasks.

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

# A   PROPERTY OF MARKOV CHAIN AND MUTUAL INFORMATION

This section lists the related properties of Markov chain and mutual information used in this work.

**Markov Chain**

Consider three random variables $\mathbf{a}$, $\mathbf{b}$ and $\mathbf{c}$ coming from a joint distribution $p(\mathbf{a}, \mathbf{b}, \mathbf{c})$, if the conditional distribution of $\mathbf{c}$ depends only on $\mathbf{a}$ and is conditionally independent of $\mathbf{b}$, the variables can form a Markov chain in the order denoted as: $\mathbf{b} \rightarrow \mathbf{a} \rightarrow \mathbf{c}$. Specially, the joint distribution can be written as (Cover, 1999):

$$p(\mathbf{a}, \mathbf{b}, \mathbf{c}) = p(\mathbf{a})p(\mathbf{b}|\mathbf{a})p(\mathbf{c}|\mathbf{a}). \tag{7}$$

Markovity implies a conditional independence between $\mathbf{c}$ and $\mathbf{b}$ when $\mathbf{a}$ is observed, the reason is:

$$p(\mathbf{c}, \mathbf{b}|\mathbf{a}) = \frac{p(\mathbf{a}, \mathbf{b}, \mathbf{c})}{p(\mathbf{a})} = \frac{p(\mathbf{c}, \mathbf{a})p(\mathbf{b}|\mathbf{a})}{p(\mathbf{a})} = p(\mathbf{c}|\mathbf{a})p(\mathbf{b}|\mathbf{a}). \tag{8}$$

By rewriting the conditional independence, the Markov chain of $\mathbf{b} \rightarrow \mathbf{a} \rightarrow \mathbf{c}$ also implies $\mathbf{c} \rightarrow \mathbf{a} \rightarrow \mathbf{b}$ as:

$$p(\mathbf{b}, \mathbf{c}|\mathbf{a}) = \frac{p(\mathbf{a}, \mathbf{b}, \mathbf{c})}{p(\mathbf{a})} = \frac{p(\mathbf{b}, \mathbf{a})p(\mathbf{c}|\mathbf{a})}{p(\mathbf{a})} = p(\mathbf{c}|\mathbf{a})p(\mathbf{b}|\mathbf{a}). \tag{9}$$

**Mutual Information**

1. Positivity: $I(\mathbf{a}; \mathbf{b}) \geq 0$; $I(\mathbf{a}; \mathbf{b}|\mathbf{c}) \geq 0$.

2. Chain Rule: $I(\mathbf{a}; \mathbf{b}, \mathbf{c}) = I(\mathbf{a}; \mathbf{b}) + I(\mathbf{a}; \mathbf{c}|\mathbf{b})$

3. $I(\mathbf{a}; \mathbf{b}) = H(\mathbf{a}) - H(\mathbf{a}|\mathbf{b}) = 0$ if and only if $\mathbf{a}$ and $\mathbf{b}$ are independent.

$$\begin{aligned}
I(\mathbf{a}; \mathbf{b}) &= H(\mathbf{a}) - H(\mathbf{a} \mid \mathbf{b}) \\
&= \mathbb{E}\left[\log \frac{1}{p(\mathbf{a})}\right] - \mathbb{E}\left[\log \frac{1}{p(\mathbf{a}|\mathbf{b})}\right] \\
&= \mathbb{E}\left[\log \frac{p(\mathbf{a}|\mathbf{b})}{p(\mathbf{a})} \frac{p(\mathbf{b})}{p(\mathbf{b})}\right] \\
&= \mathbb{E}\left[\log \frac{p(\mathbf{a}, \mathbf{b})}{p(\mathbf{a})\mathbf{b}}\right] \\
&= D\left(p(\mathbf{a}, \mathbf{b}) \| p(\mathbf{a})) \times p(\mathbf{b})\right) \\
&\geq 0
\end{aligned} \tag{10}$$

4. Data-Processing Inequality (DPI): If three random variables $\mathbf{a}$, $\mathbf{b}$ and $\mathbf{c}$ coming from a joint distribution $p(\mathbf{a}, \mathbf{b}, \mathbf{c})$ can form a Markov chain in the order denoted as: $\mathbf{b} \rightarrow \mathbf{a} \rightarrow \mathbf{c}$, then $I(\mathbf{b}; \mathbf{a}) \geq I(\mathbf{b}; \mathbf{c})$. (Theorem 2.8.1 in Cover (1999))

# B   LEARNING OBJECTIVES DECOMPOSITION

In this section, we decompose the proposed learning objective to analyze the relationship of learning objectives between Equation 1 and $\beta$-VAE-based methods. Let $p(\mathbf{x})$ denote the true distribution of the data, and $p_\phi(\mathbf{z}|\mathbf{x})$ and $p_\theta(\mathbf{x}|\mathbf{z})$ denote the unknown distributions that we need to estimate, parametrized by an encoder with $\phi$ and a decoder with $\theta$.

## B.1   DECOMPOSITION OF MINIMAL SUFFICIENCY

*Minimal Sufficiency* is defined as $\mathbf{z}$ can encode the minimum amount information of $\mathbf{x}$ required to reconstruct $\hat{\mathbf{x}}$, optimized by $minimize\ I(\mathbf{x}; \mathbf{z})$. Inspired from the work of Chen et al. (2018), we can

decompose the minimal sufficiency by assuming the prior $p(\mathbf{z})$ as a factorized Gaussian as:

$$
\begin{aligned}
I(\mathbf{x};\mathbf{z}) &= \mathbb{E}_{q(\mathbf{x},\mathbf{z})}\left[\mathrm{KL}\left(q(\mathbf{x},\mathbf{z})\,\|\,q(\mathbf{z})\,p(\mathbf{x})\right)\right] \\
&= \mathbb{E}_{q(\mathbf{x},\mathbf{z})}\left[\log\frac{q(\mathbf{x},\mathbf{z})}{q(\mathbf{z})\,p(\mathbf{x})}\right] = \mathbb{E}_{q(\mathbf{x},\mathbf{z})}\left[\log\frac{q(\mathbf{z}|\mathbf{x})\,p(\mathbf{x})}{q(\mathbf{z})\,p(\mathbf{x})}\right] \\
&= \mathbb{E}_{p(\mathbf{x})}\left[\mathbb{E}_{q(\mathbf{z}|\mathbf{x})}\left[\log q(\mathbf{z}|\mathbf{x}) - \log q(\mathbf{z}) + \log p(\mathbf{z}) - \log p(\mathbf{z}) + \log\prod_j q(\mathbf{z}_j) - \log\prod_j q(\mathbf{z}_j)\right]\right] \\
&= \mathbb{E}_{q(\mathbf{x},\mathbf{z})}\left[\log\frac{q(\mathbf{z}|\mathbf{x})}{q(\mathbf{z})}\right] - \mathbb{E}_{q(\mathbf{z})}\left[\sum_j \log\frac{q(\mathbf{z}_j)}{p(\mathbf{z}_j)}\right] - \mathbb{E}_{q(\mathbf{z})}\left[\log\frac{q(\mathbf{z})}{\prod_j q(\mathbf{z}_j)}\right] \\
&= \mathbb{E}_{p(\mathbf{x})}\left[\mathrm{KL}\left(q(\mathbf{z}|\mathbf{x})\,\|\,p(\mathbf{z})\right)\right] - \sum_j \mathrm{KL}\left(q(\mathbf{z}_j)\,\|\,p(\mathbf{z}_j)\right) - \mathrm{KL}\left(q(\mathbf{z})\,\Big\|\,\prod_j q(\mathbf{z}_j)\right)
\end{aligned}
\tag{11}
$$

The first term is KL divergence between inferred posterior and prior, usually as a penalty term of $\beta$-VAEs for disentangling. The second term is the dimension-wise KL divergence, which represents the distance between each latent dimension to the prior. The third term is the Total Correlation referring to the independence among variables.

## B.2 ESTIMATION OF DISENTANGLEMENT

*Disentanglement* is defined as the independence between any two variables, optimized by minimizing $\sum_{i\neq j} I(\mathbf{z}_i;\mathbf{z}_j)$, which is an lower bound of Total Correlation.

The concept of Total Correlation (TC) was described by McGill (1954) and formally formulated by Watanabe (1960) to evaluate the mutual independence of multi-variant variables. Assume a set of $\mathbf{z}$ of random variables $\mathbf{z}_1, \ldots, \mathbf{z}_n$. The TC in $\mathbf{z}$ is expressed as:

$$
C(\mathbf{z}_1, \ldots, \mathbf{z}_n) = \sum_{i=1}^{n} H(\mathbf{z}_i) - H(\mathbf{z}_1, \mathbf{z}_2, \ldots, \mathbf{z}_n),
\tag{12}
$$

where $H(\mathbf{z}_1, \mathbf{z}_2, \ldots, \mathbf{z}_n)$ denotes the joint entropy. Furthermore, according to the work of Te Sun (1980; 1975), the relation between TC and mutual information can be described as:

$$
C(\mathbf{z}) = \sum_{i=1}^{n} H(z_i) - H(\mathbf{z}) = -\sum_{n\geq 2}\Delta H(\mathbf{z})
\tag{13}
$$

The general $\Delta H(\mathbf{z})$ is defined as Fano's multiple mutual information among variables. In the case of $n=2$, it reduces to Shannon's mutual information and in the case of $n=3$, it coincides with McGill's mutual information. The formulation of TC can be simplified as:

$$
C(\mathbf{z}_1, \ldots, \mathbf{z}_n) = \sum_{j\neq i} I(\mathbf{z}_i;\mathbf{z}_j) + \sum_{k\neq j\neq i}\Delta H(\mathbf{z}_i;\mathbf{z}_j;\mathbf{z}_k) + \ldots + \Delta H(\mathbf{z}_1;\ldots;\mathbf{z}_n)
\tag{14}
$$

In VAEs, to measure dependence for multiple latent variables, TC is computed as $D_{KL}\left(q(\mathbf{z})\,\|\,\prod_j q(\mathbf{z}_j)\right)$. Therefore, the proposed disentanglement $\sum_{i\neq j} I(\mathbf{z}_i;\mathbf{z}_j)$ is a lower bound of TC.

### B.3 RELATION WITH OTHER WORKS

After decomposing the minimal sufficiency and estimating the disentanglement by TC, we can combine two regularizers as:

$$\lambda_1 I\left(\mathbf{x};\mathbf{z}\right) + \lambda_2 \sum_{j \neq i} I\left(\mathbf{z}_i;\mathbf{z}_j\right)$$

$$\leq \lambda_1 \mathbb{E}_{p(\mathbf{x})}\left[\mathrm{KL}\left(q\left(\mathbf{z}|\mathbf{x}\right)\|p\left(\mathbf{z}\right)\right)\right] - \lambda_{12} \sum_j \mathrm{KL}\left(q\left(\mathbf{z}_j\right)\|p\left(\mathbf{z}_j\right)\right) - \lambda_{13}\,\mathrm{KL}\left(q\left(\mathbf{z}\right)\left\|\prod_j q\left(\mathbf{z}_j\right)\right.\right)$$

$$+ \lambda_2 \sum_j \mathrm{KL}\left(q\left(\mathbf{z}_j\right)\|p\left(\mathbf{z}_j\right)\right)$$

$$= \lambda_1 \mathbb{E}_{p(\mathbf{x})}\left[\mathrm{KL}\left(q\left(\mathbf{z}|\mathbf{x}\right)\|p\left(\mathbf{z}\right)\right)\right] + \left(\lambda_2 - \lambda_{12}\right) \sum_j \mathrm{KL}\left(q\left(\mathbf{z}_j\right)\|p\left(\mathbf{z}_j\right)\right) - \lambda_{13}\,\mathrm{KL}\left(q\left(\mathbf{z}\right)\left\|\prod_j q\left(\mathbf{z}_j\right)\right.\right)$$

$$= \lambda_a \mathbb{E}_{p(\mathbf{x})}\left[\mathrm{KL}\left(q\left(\mathbf{z}|\mathbf{x}\right)\|p\left(\mathbf{z}\right)\right)\right] + \lambda_b \sum_j \mathrm{KL}\left(q\left(\mathbf{z}_j\right)\|p\left(\mathbf{z}_j\right)\right) + \lambda_c\,\mathrm{KL}\left(q\left(\mathbf{z}\right)\left\|\prod_j q\left(\mathbf{z}_j\right)\right.\right)$$

(15)

Therefore, by assigning different weights to the decomposed term, we can establish the relationship between the proposed information-theoretic objective and existing $\beta$-VAE variants, which will provide insights regarding the capabilities and limitations of existing methods, and further motivate the proposed work.

Specially, the objective function of $\beta$-TCVAE is $I\left(\mathbf{x},\mathbf{z}\right) + KL\left(q\left(\mathbf{z}\right)\|\prod_j q\left(\mathbf{z}_j\right)\right) + \sum_j KL\left(q\left(\mathbf{z}_j\right)\|p\left(\mathbf{z}_j\right)\right)$. If we decompose the $I\left(\mathbf{x};\mathbf{z}\right)$ of $\beta$-TCVAE as: $KL\left(q\left(\mathbf{z}|\mathbf{x}\right)\|p\left(\mathbf{z}\right)\right) - KL\left(q\left(\mathbf{z}\right)\|\prod_j q\left(\mathbf{z}_j\right)\right) - \sum_j KL\left(q\left(\mathbf{z}_j\right)\|p\left(\mathbf{z}_j\right)\right)$, the regularizer term of $\beta$-TCVAE can be written as: $\alpha KL\left(q\left(\mathbf{z}|\mathbf{x}\right)\|p\left(\mathbf{z}\right)\right) + \left(\beta - \alpha\right) KL\left(q\left(\mathbf{z}\right)\|\prod_j q\left(\mathbf{z}_j\right)\right) + \left(\gamma - \alpha\right) \sum_j KL\left(q\left(\mathbf{z}_j\right)\|p\left(\mathbf{z}_j\right)\right)$. Since $\alpha = \gamma = 1$ in $\beta$-TCVAE, therefore when $\lambda_c = 0$, the last line of Equation 15 is equivalent to $\beta$-TCVAE.

## C PROOF OF COMPOSITIONAL MINIMAL SUFFICIENCY

In this section, we prove the statements reported in the section 2.2 of the paper.

*Property:*

$(P_1)$: Data-processing inequality in the markov chain: considering the markov chain $\mathbf{b} \to \mathbf{a} \to \mathbf{c}$, then $I\left(\mathbf{b},\mathbf{a}\right) \geq I\left(\mathbf{b};\mathbf{c}\right)$. (Theorem 2.8.1 (Cover, 1999).)

$(P_2)$: Chain rule for mutual information: $I\left(\mathbf{a};\mathbf{b},\mathbf{c}\right) = I\left(\mathbf{a};\mathbf{b}\right) + I\left(\mathbf{a};\mathbf{c}|\mathbf{b}\right)$. (Theorem 2.5.1 (Cover, 1999).)

$(P_3)$: Decompositon of conditional mutual information. (Proposition B.1. (Federici et al., 2020).)

*Conditional independence assumptions:*

The Markov Chain $\mathbf{m}_j^{l+1} \to \mathbf{x} \to \left(\mathbf{m}_i^l, \mathbf{m}_j^l\right)$, implies the conditional independence between $\mathbf{m}_j^{l+1}$ and $\left(\mathbf{m}_i^l, \mathbf{m}_j^l\right)$ when $\mathbf{x}$ is observed.

**Proof**:

$$I\left(\mathbf{x};\mathbf{m}_j^{l+1}\right) \overset{(P_1)}{\geq} I\left(\mathbf{m}_j^{l+1};\mathbf{m}_i^l,\mathbf{m}_j^l\right)$$
$$\overset{(P_2)}{=} I\left(\mathbf{m}_j^{l+1};\mathbf{m}_j^l\right) + I\left(\mathbf{m}_j^{l+1};\mathbf{m}_i^l \mid \mathbf{m}_j^l\right)$$
$$\overset{(P_3)}{=} I\left(\mathbf{m}_j^{l+1};\mathbf{m}_j^l\right) + I\left(\mathbf{m}_j^{l+1};\mathbf{m}_i^l\right) - I\left(\mathbf{m}_i^l;\mathbf{m}_j^l\right).$$

(16)

## D    EXPERIMENTAL DETAILS

For all baselines, we use a Convolutional Neural Network for the encoder, and a Deconvolutional Neural Network for the decoder. Specially, for Factor-VAE, we use a 6-layer Multi-Layer Perceptron for the discriminator, with the leaky ReLU as the activation on per layer, and we set $\gamma$ as 6.4. For $\beta$-VAE, we use a set of $\beta = \{4, 16, 60\}$. Annealed-VAE uses $\gamma = 1000$ with a linearly increasing $C$ from 0.5 nats to a 25.0 nats). $\beta$-TCVAE uses $\alpha = \gamma = 1$ and $\beta = 4$. DIP-VAE is implemented by the parameters $\lambda_{od} = 100$ and $\lambda_d = 10$.

For RecurD, we implement $RecurD\ 0$ as the same encoder/decoder architecture with all baselines, as shown in Table 3. As for $RecurD\ 1$ and $RecurD\ 2$, we implement the same decoder architecture with all baselines, and the details about the encoder of $RecurD\ 1$ and $RecurD\ 2$ are shown in Table 4 and Table 5. Following the previous work, we use negative cross-entropy to compute reconstruction error, which represents sufficiency in our work. As for the computation of minimal sufficiency and disentanglement, we implement the MINE estimator by 4-layer Multi-Layer Perceptron, similar with the work of Federici et al. (2020). As for the hyperparameter in our model, we vary $\lambda_1$ and $\lambda_2$ in the set $\{0.1, 0.2, 0.5, 1, 2, 5, 10, 50\}$ while fixing $\lambda_1 = 1$ and $\lambda_1 = 2$ for both datasets. During training, we use Adam optimiser with learning rate $1e - 4$, $\beta_1 = 0.9$, $\beta_2 = 0.999$ for parameter updates. Specially, we utilize RecurD 1 on dSprites, 3DCars and 3DShapes, and RecurD 2 on CelebA.

Table 3: Encoder and Decoder architecture for all baselines and $RecurD\ 0$.

| Input | dSprites (3DShapes): $64 \times 64 \times 1(3)$ Images |
|---|---|
| | Conv: k=4, s=2, p=1, channel=32, ReLU |
| | Conv: k=4, s=2, p=1, channel=32, ReLU |
| Encoder | Conv: k=4, s=2, p=1, channel=64, ReLU |
| | Conv: k=4, s=2, p=1, channel=64, ReLU |
| | FC: 128 (256) |
| | FC: $2 \times 8$ |
| | Input: $\mathbf{z} \in \mathbb{R}^8$ |
| | FC: 128, ReLU |
| | FC: $4 \times 4 \times 64$, ReLU |
| Decoder | Deconv: k=4, s=2, p=1, channel=64, ReLU |
| | Deconv: k=4, s=2, p=1, channel=32, ReLU |
| | Deconv: k=4, s=2, p=1, channel=32, ReLU |
| | Deconv: k=4, s=2, p=1, channel=1(3), ReLU |
| Output: | $64 \times 64 \times 1(3)$ Reconst. Images |

Table 4: Encoder architecture of $RecurD\ 1$.

| Input | | $64 \times 64 \times 1(3)$ Image |
|---|---|---|
| 0-th Recursive Module | Router(**v**) | - |
| | GoE | $4\times$ [Conv: k=4, s=2, p=1, channel=32, ReLU] |
| | | Conv: k=4, s=2, p=1, channel=64, ReLU |
| 1-th Recursive Module | Router(**v**) | FC, 8, ReLU |
| | GoE | $8 \times \begin{bmatrix} \text{Conv : k=4, s=2, p=1, channel=16, ReLU} \\ \text{FC, 16; FC, } 2 \times 1 \end{bmatrix}$ |

Table 5: Encoder architecture of $RecurD$ 2.

| Input | | $64 \times 64 \times 1$ (3) Image |
|---|---|---|
| 0-th Recursive Module | Router($\mathbf{v}$) | - |
| | GoE | $2\times$ [Conv: k=4, s=2, p=1, channel=32, ReLU ] |
| 1-th Recursive Module | Router($\mathbf{v}$) | FC, 4, ReLU |
| | GoE | $4 \times$ [Conv: k=4, s=2, p=1, channel=16, ReLU] |
| 2-th Recursive Module | Router($\mathbf{v}$) | FC, 16, ReLU |
| | GoE | $16 \times \begin{bmatrix} \text{Conv}: \text{k=4, s=2, p=1, channel=16, ReLU} \\ \text{FC}, 16; \text{FC}, 2 \times 1 \end{bmatrix}$ |

# E  ADDITIONAL EXPERIMENTS

This section reports additional experiments, which draw similar conclusions as that of Section 5.

## E.1  QUANTITATIVE RESULTS

The first set of quantitative results is the supplementary results of disentanglement score DCI-C and DCI-I on dSprites and 3DShapes. Table 6, Figure 8 and Figure 9 show the results of disentanglement score with reconstruction error, which also show RecurD can achieve a better trade-off between reconstruction and disentanglement.

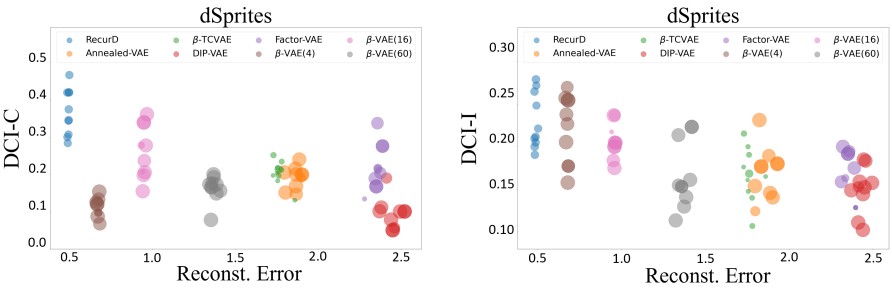

Figure 8: Reconstruction error vs. disentanglement performance on dSprites. Scatters located at the left top indicate better performance.

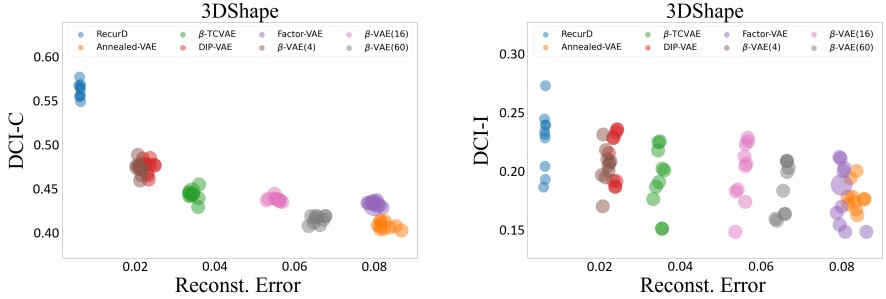

Figure 9: Reconstruction error vs. disentanglement performance on 3DShapes. Scatters located at the left top indicate better performance.

The second set is the performance comparison results with three additional baselines, i.e, Info-GAN (Chen et al., 2016), Control-VAE and NVAE. InfoGAN (Chen et al., 2016) maximizes the mutual information between the small subset of the latent variables and the observations to increase the interpretability of the latent representation. Control-VAE (Shao et al., 2020) dynamically tunes

Table 6: DCI-C, DCI-I scores and reconstruction error on the test sets for dSprites and 3DShapes. Bold face indicates the best results, i.e., reconstruction error and disentanglement scores.

| dataset | dSprites | | | 3DShapes | | |
|---|---|---|---|---|---|---|
| | Reconst error | DCI-C | DCI-I | Reconst error | DCI-C | DCI-I |
| $\beta$-VAE($\beta = 4$) | 0.0066 | 0.1148 | 0.2181 | 0.0216 | 0.4463 | 0.1982 |
| $\beta$-VAE($\beta = 16$) | 0.0094 | 0.1575 | 0.2087 | 0.0544 | 0.4691 | 0.1828 |
| $\beta$-VAE($\beta = 60$) | 0.0127 | 0.1261 | 0.1769 | 0.0624 | 0.4215 | 0.1641 |
| Annealed-VAE | 0.0171 | 0.1793 | 0.1663 | 0.0811 | 0.4419 | 0.1682 |
| Factor-VAE | 0.0228 | 0.1375 | 0.1699 | 0.0800 | 0.4419 | 0.1707 |
| $\beta$-TCVAE | 0.0162 | 0.1988 | 0.1660 | 0.0312 | 0.4516 | 0.1774 |
| DIP-VAE | 0.0213 | 0.0968 | 0.1496 | 0.0213 | 0.4621 | 0.1961 |
| RecurD | **0.0047** | **0.3835** | **0.2222** | **0.0083** | **0.5668** | **0.2014** |

the weight $\beta$ on the KL term to achieve a good trade-off between disentanglement and reconstruction quality. NVAE (Vahdat & Kautz, 2020) optimizes high-quality image generation via global correlation capturing across multi-layer latent variables.

We compare with Info-GAN Control-VAE and NVAE on three datasets, including dSprites, 3Dshapes and 3Dcars. The 3DCars (Reed et al., 2015) exhibits different car models from Fidler et al. (2012) under different camera viewpoints. The evaluation method is the disentanglement score MIG. Table 7 show that NVAE and VampPrior can achieve comparable reconstruction error compared to RecurD. However, their disentanglement qualities are not as good as RecurD, because RecurD additionally regularizes the inter-layer information sharing and alleviates the information redundancy of multiple layers. This experiment demonstrates the superiority of the proposed RecurD method in disentanglement compared to existing hierarchical VAEs.

Table 7: MIG score and reconstruction error on the test sets for dSprites, 3DShapes and 3DCars. Bold face indicates the best results, i.e., reconstruction error and disentanglement scores.

| dataset | dSprites | | 3DShapes | | 3DCars | |
|---|---|---|---|---|---|---|
| | Reconst. Error | MIG | Reconst. Error | MIG | Reconst. Error | MIG |
| $\beta$-VAE($\beta = 4$) | 0.0066 | 0.2617 | 0.0216 | 0.2519 | 0.0376 | 0.1015 |
| InfoGAN | - | 0.1598 | - | 0.1874 | - | 0.1083 |
| Control-VAE | 0.0102 | 0.2455 | 0.0357 | 0.2630 | 0.0257 | 0.1583 |
| NVAE | 0.0041 | 0.0043 | 0.0078 | 0.0081 | 0.0118 | 0.0034 |
| RecurD | **0.0047** | **0.2707** | **0.0083** | **0.3105** | **0.0132** | **0.1762** |

## E.2 ABLATION STUDY

In this section, we report the supplementary results of ablation study. The first is to study the impact of compositional learning objective with varying regularization coefficients ($\lambda_1$ and $\lambda_2$) of minimal sufficiency and disentanglement. Figure E.2 and Figure E.2 show the scatter plots of disentanglement scores along with reconstruction error on the test sets of dSprites, with varying $\lambda_1$, $\lambda_2$. Figure 12 and Figure 13 show the results on the 3DShapes.

The second set is to evaluate the influence of hyperparameter $k$ — the group size in the GoE of Recursive Module. Figure E.2 and Figure E.2 show the influence of varying $k$ for disentanglement scores and reconstruction error on the test sets of dSprites. Figure E.2 and Figure E.2 show the results on the 3DShapes.

The third set is to evaluate the disentanglement metrics on the preceding layers. We compute MIG on $m^{L-1}$ and $m^{L-2}$ of RecurD on 3DShapes. As shown in Table 8, high-level representations have higher MIG than that of low-level representations ($z > m^{L-1} > m^{L-2}$). These results confirm that

the recursive propagation of inductive bias through the feed-forward network improves disentangled representation learning.

The last set is to evaluate the impact of architecture of Gate-of-Encoders (GoE), we conduct three variant GoEs, including **Linear**: GoE is implemented as a linear layer with softmax; **Fix**: GoE is implemented as fix assignment, i.e., we split $m_l$ in $d+1$ equal slices; and **Att**: GoE is implemented as a multi-head attention layer (the head is fixed as $8$). As shown in Table 9, this study demonstrates that GoE indeed fits well with the disentanglement process.

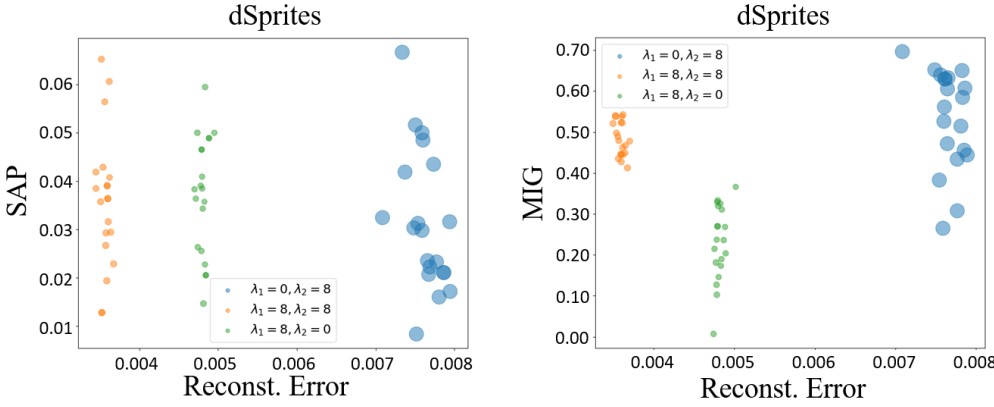

Figure 10: Ablation study on $\lambda_1$ and $\lambda_2$ on dSprites.

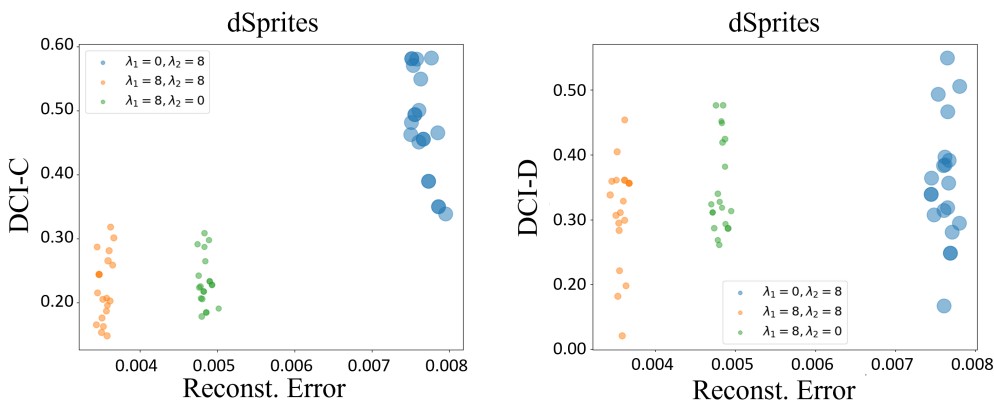

Figure 11: Ablation study on $\lambda_1$ and $\lambda_2$ on dSprites.

Table 8: Comparison of MIG on $z$, $m^{L-1}$ and $m^{L-2}$.

|  | $m^{L-2}$ | $m^{L-1}$ | $z(m^L)$ | Reconst. Error |
|---|---|---|---|---|
| MIG | 0.1081 | 0.2892 | 0.3105 | 0.0083 |

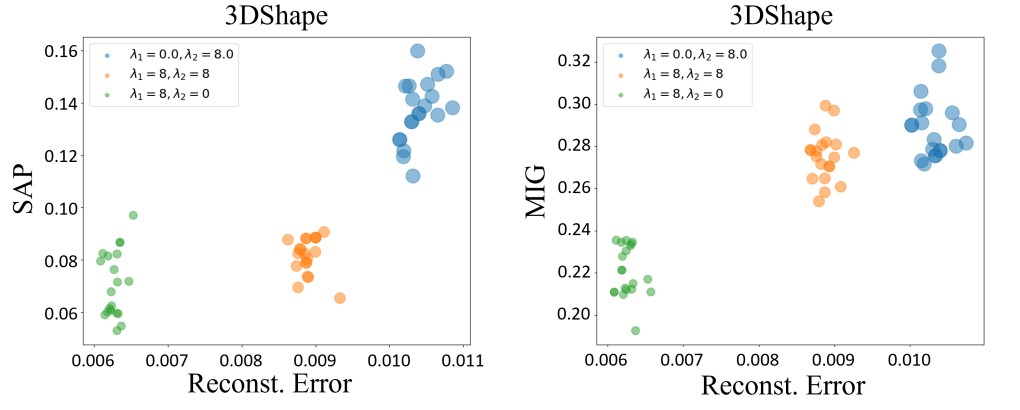

Figure 12: Ablation study on $\lambda_1$ and $\lambda_2$ on 3DShapes.

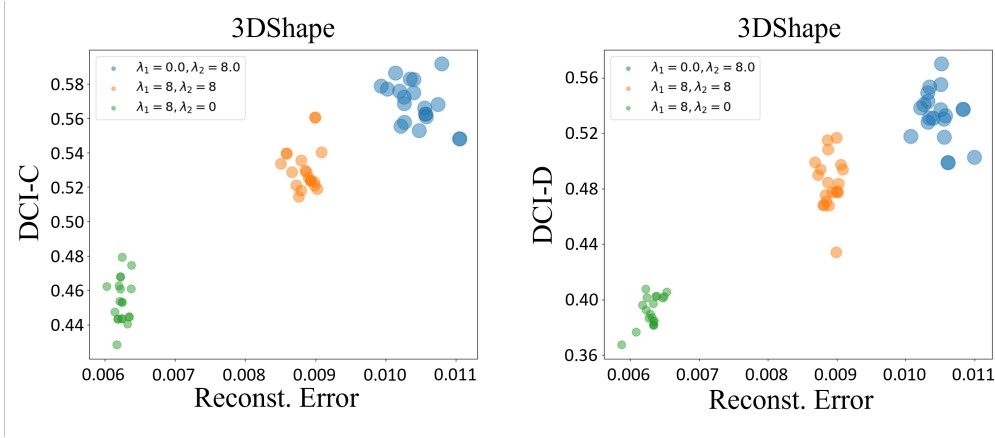

Figure 13: Ablation study on $\lambda_1$ and $\lambda_2$ on 3DShapes.

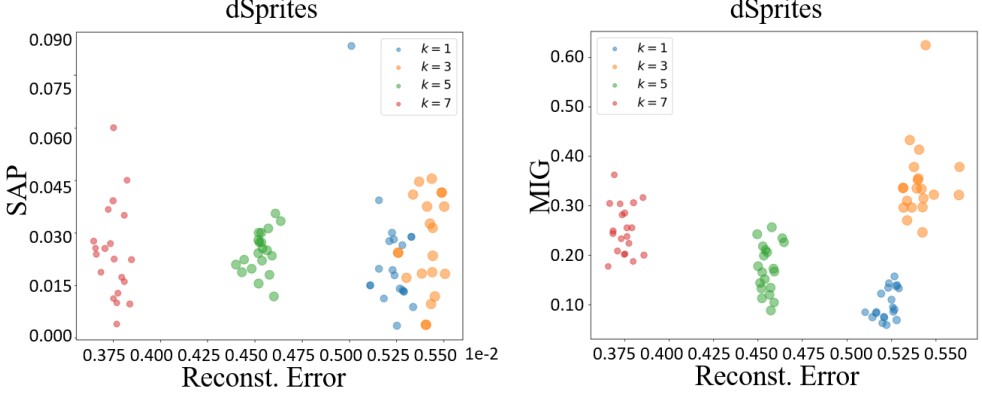

Figure 14: Ablation study on $k$ on dSprites.

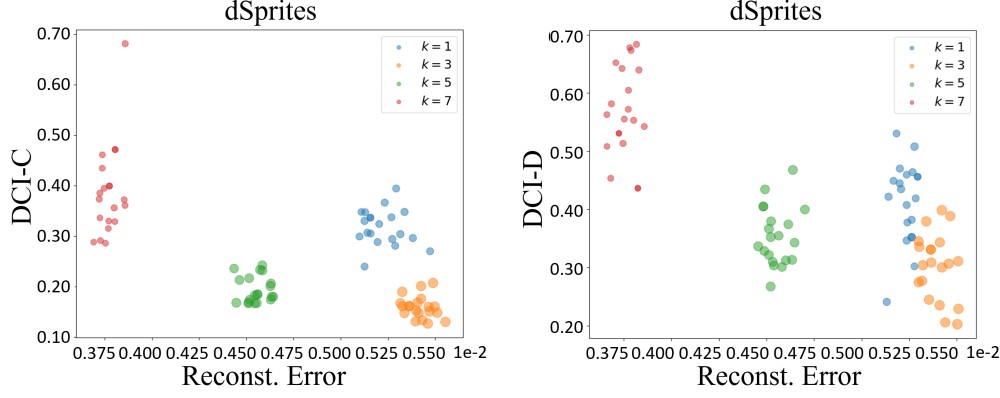

Figure 15: Ablation study on $k$ on dSprites.

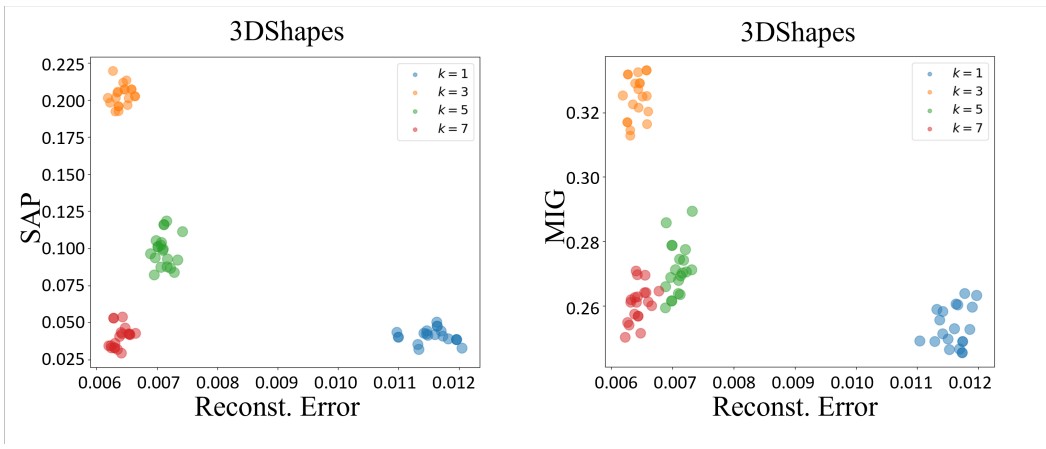

Figure 16: Ablation study on $k$ on 3DShapes.

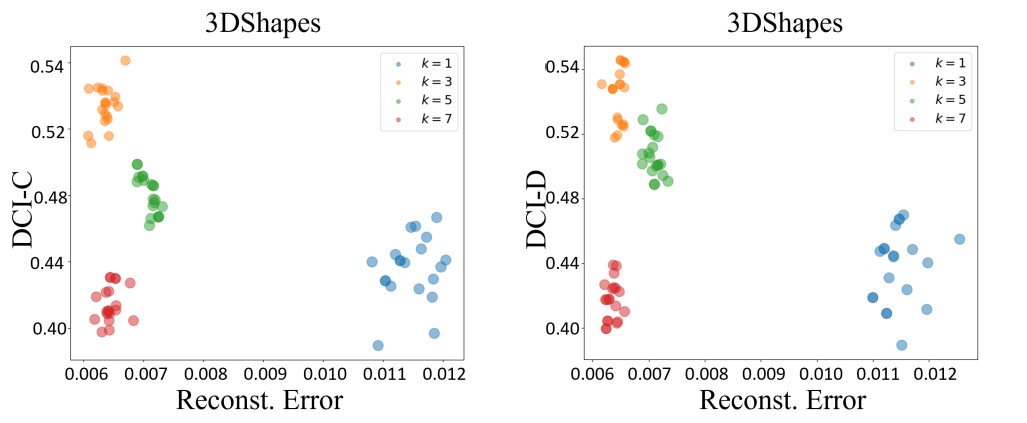

Figure 17: Ablation study on $k$ on 3DShapes.

Table 9: Ablation study on the GoE architecture.

|  | Linear | Fix | Att | RecurD |
|---|---|---|---|---|
| MIG | 0.2539 | 0.2617 | 0.2604 | 0.3105 |
| Reconst. Error | 0.0176 | 0.0116 | 0.0095 | 0.0083 |

### E.3 QUALITATIVE RESULTS

In this section, we report the qualitative samples of traversal images on three datasets, including dSprites(Figure 18), 3DShapes(Figure 19) and CelebA(from Figure 20 to Figure 25).

Specifically, on CelebA, we tentatively increase the dimensionality of latent variables of RecurD (from 16 to 32) by doubling the output of the last layer of each encoder from a single latent variable to a 2-dimensional variable.

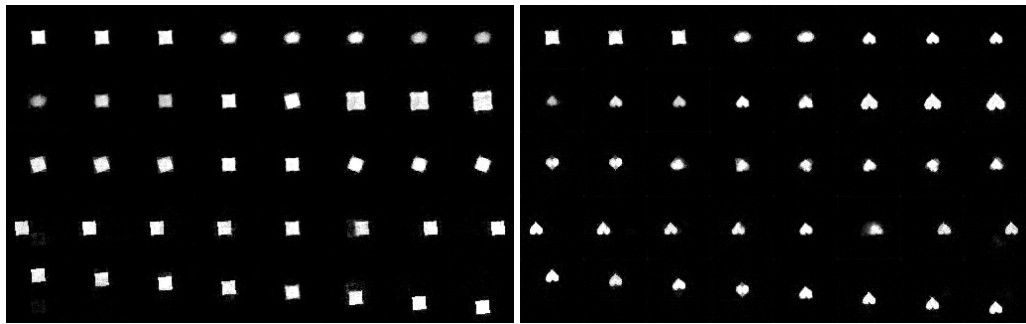

Figure 18: Traversal samples on dSprites.

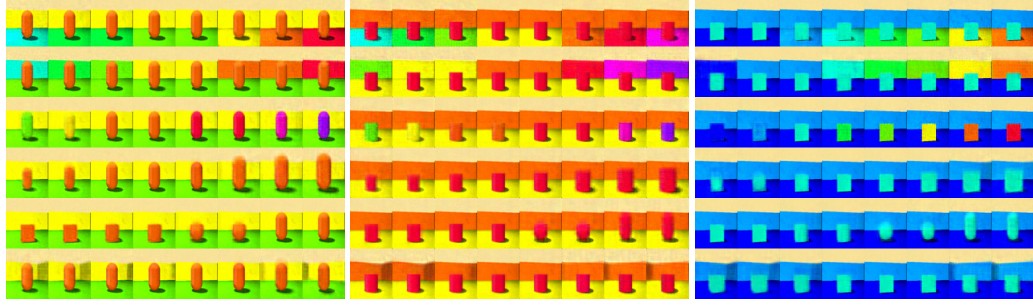

Figure 19: Traversal samples on 3DShapes.

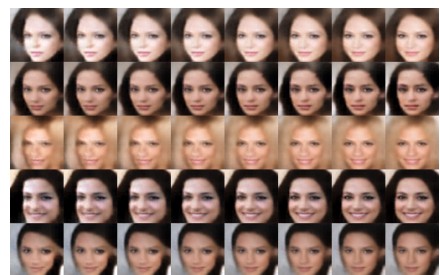

Figure 20: Traversal samples on CelebA (Azimuth).

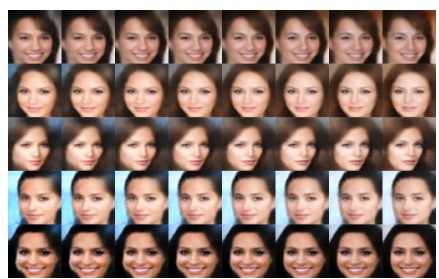

Figure 21: Traversal samples on CelebA (Background Color).

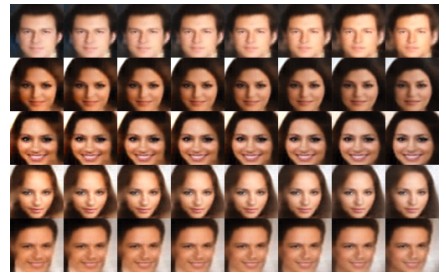

Figure 22: Traversal samples on CelebA (Face Color).

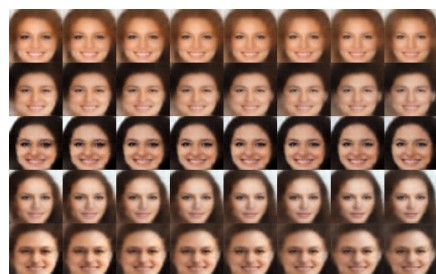

Figure 23: Traversal samples on CelebA (Face Width).

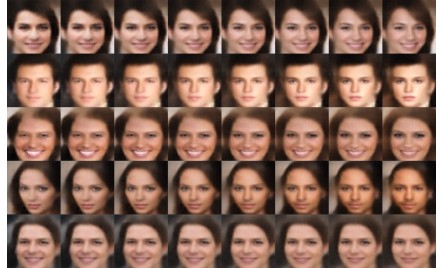

Figure 24: Traversal samples on CelebA (Gender).

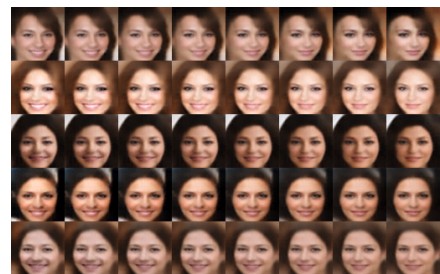

Figure 25: Traversal samples on CelebA (Smile).

### E.4 EVALUATION OF COMPUTATIONAL COSTS

In this section, we evaluate the computational complexity of our model on 3DShapes and CelebA. The evaluation metrics consist of multiply-accumulate operation (MACs), the model parameters (Params), evaluation time (Eva. time) and converge time (all models converge to the same reconstruction error as $\beta$-VAE, denoted as Con. time).

In terms of parameter efficiency, as shown in Table 10, the recursive disentanglement network itself only contains 0.826 million parameters (RecurD 2 w/o MINEs), which are comparable to Beta-VAE. Most of the parameters of RecurD 2 are contributed by using MINE estimator. Specifically, in the initial implementation (RecurD 2/specific MINEs), each pair of outputs of encoders is equipped with a specific MINE model, which contributes 7.214 million parameters. We further optimize the design by using shared MINE within the same feature category, which can effectively reduce the total number of parameters down to 1.325 million (RecurD 2/shared MINEs).

Table 10: Complexity comparison of three models on 3DShapes and CelebA.

| Dataset | Method | MACs(G) | Params(M) | Eva. time(s) | Con. time(s) |
|---------|--------|---------|-----------|--------------|--------------|
| 3DShapes | RecurD 1 | 3.466 | 3.672 | 0.005124 | 27.1212 |
| | RecurD 2 | 3.469 | 3.694 | 0.005354 | 23.3197 |
| | beta-VAE | 3.144 | 0.769 | 0.003283 | 32.5251 |
| | Factor-VAE | 3.401 | 4.779 | 0.003389 | 40.6231 |
| CelebA | RecurD 1 | 3.944 | 7.935 | 0.01065 | 29.7421 |
| | RecurD 2 | 3.957 | 8.040 | 0.01087 | 36.6408 |
| | RecurD 2 w/o MINEs | 2.960 | 0.826 | - | - |
| | RecurD 2/shared MINEs | 3.654 | 1.325 | 0.01072 | 36.6402 |
| | beta-VAE | 3.145 | 0.769 | 0.01030 | 45.2067 |
| | Factor-VAE | 3.402 | 4.792 | 0.01005 | 48.9939 |

