# OpenReview forum: "Recursive Disentanglement Network"
_ICLR.cc/2022/Conference — ICLR 2022 Poster_

### Official Review · Reviewer_p3Zz · 2021-10-24

**Correctness:** 3
**Technical Novelty And Significance:** 3
**Empirical Novelty And Significance:** 3
**Recommendation:** 6
**Confidence:** 5

**Main Review:**

Pros:
1. Developed a compositional disentanglement learning, called RecurD, that directs the disentanglement learning process across the compositional feature space.
2. Provide some theoretical analysis based on information theory.

Cons:
1. Optimizing the lower bound of Eq.2 does not mean obtaining the optimal objective function of  $\beta$-TCVAE paper [1] on the right-hand side of Eq 2. As far as I know, the right-hand side of Eq 2 is the objective of $\beta$-TCVAE. If we optimize the objective function on the left-hand side, it does not hold for optimizing $\beta$-TCVAE. Thus, I am afraid that the proposed objective function in Eq 1 fails to be generalized to the existing $\beta$-TCVAE and FactorVAE. In contrast, optimizing the objective of $\beta$-TCVAE is approximately equivalent to optimizing the proposed objective function in this paper. What is more, in Table 1, $\lambda_c =1$ for the original $\beta$-TCVAE in their paper.
2. Do not specify the number of Gate of Encoders (GoE) for different datasets. It is hard to know how many GoE should be used for a new dataset, like CelbeA, that contains 40 latent variables. Also, I am curious about the complexity of the proposed network?
3. The upper bound and lower bound are confounding. On page 9, the author said: mutual information I(x,z) is the upper bound of KL divergence. In fact, based on the proof in prior work [2], I(x,z) is the lower bound of KL divergence.
4. The Markov Chain in Eq 7 is incorrect. Based on my understanding, the next state $X_{t+1}$ of Markov chain is only related to the current state $X_t$. Hence the joint probability of p(a,b,c) = p(a)p(b|a)p(c|a) rather than p(a)p(b|a)p(c|b). This is because c is conditionally independent on b as you mentioned, b->a->c.
5. Missing baselines in recent years. The author only discussed and compare the proposed method with the baselines before the year of 2019. There are some recent works [3] [4] [5] on improving the disentanglement and reconstruction error. For instance, ControlVAE [5] dynamically tunes the weight \beta on the KL term to achieve a good trade-off between disentanglement and reconstruction error.
6. Did not conduct experiments on complex datasets. The authors should do experiments on 3D chairs or CelebA to demonstrate the good performance of the proposed method.
7. The result in Fig 5 does not perform well. We can observe that for orientation and scale factors, they are slightly entangled. Besides, the reconstruction quality is not as good as ControlVAE and FactorVAE in the paper. In particular, ControlVAE and FactorVAE  can disentangle both 5 latent factors which are better than those in this work.
8. There are some typos in this paper, please proofread this manuscript. For instance, priopr work --> prior work on Page 5.

Reference:
[1] Chen, R. T., Li, X., Grosse, R., & Duvenaud, D. (2018). Isolating sources of disentanglement in variational autoencoders. arXiv preprint arXiv:1802.04942.
[2] Xue Bin Peng, Angjoo Kanazawa, Sam Toyer, Pieter Abbeel, Sergey Levine: Variational Discriminator Bottleneck: Improving Imitation Learning, Inverse RL, and GANs by Constraining Information Flow. ICLR 2019.
[3] Patrick Esser, Johannes Haux, Bj rn Ommer: Unsupervised Robust Disentangling of Latent Characteristics for Image Synthesis. ICCV 2019.
[4] Srivastava, Akash, Yamini Bansal, Yukun Ding, Cole Hurwitz, Kai Xu, Bernhard Egger, Prasanna Sattigeri, Josh Tenenbaum, David D. Cox, and Dan Gutfreund. "Improving the Reconstruction of Disentangled Representation Learners via Multi-Stage Modelling." arXiv preprint arXiv:2010.13187 (2020).
[5] Shao, H., Yao, S., Sun, D., Zhang, A., Liu, S., Liu, D., ... & Abdelzaher, T. (2020, November). Controlvae: Controllable variational autoencoder. In International Conference on Machine Learning (pp. 8655-8664). PMLR.



**Summary Of The Paper:**

This paper proposed a recursive disentanglement network (RecurD) for the learning of disentangled representations from information theoretic perspective. The experimental results show RecurD outperforms some existing baselines on two benchmark datasets.

**Summary Of The Review:**

I think the authors have addressed most of my concerns. I will increase the final rate.

---

> ### Author Response · Authors · 2021-11-17
> **Response to Reviewer p3Zz (Part 1/3)**
>
> > **Q1:** Optimizing the lower bound of Eq.2 does not mean obtaining the optimal objective function of $\beta$-TCVAE paper [1] on the right-hand side of Eq 2. As far as I know, the right-hand side of Eq 2 is the objective of $\beta$-TCVAE. If we optimize the objective function on the left-hand side, it does not hold for optimizing $\beta$-TCVAE. Thus, I am afraid that the proposed objective function in Eq 1 fails to be generalized to the existing $\beta$-TCVAE and FactorVAE. In contrast, optimizing the objective of $\beta$-TCVAE is approximately equivalent to optimizing the proposed objective function in this paper. What is more, in Table 1, $\lambda_c=1$ for the original $\beta$-TCVAE in their paper.
>
> **A1:** Thanks for your comment. The  explanations about Eqn. 2 were not detailed enough due to the space limitation. Here, we would like to add more clarifications about your comment that "optimizing the objective of $\beta$-TCVAE is approximately equivalent to optimizing the proposed objective function in this paper". And these clarifications have also been included in the Appendix B.3 of the revised paper.
>
> First, the right-hand side of Eq.2 is the upper bound of the objective of the $\beta$-TCVAE, since the $KL(q(z|x)\|p(z))$ (in Eq.2) is the upper bound of $I(x;z)$ (in $\beta$-TCVAE), see also [5]. And when $\lambda_a=1$ and $\lambda_b=\beta$, the right-hand side of Eq.2 reduce to the optimal learning objective of $\beta$-TCVAE. (* note, please see blow)
> Second, Our learning objective is to minimize Eq.1 and the right-hand of Eq.2 is the upper bound, thus minimizing the upper bound can be regarded as a surrogate object as our approach. The purpose of the decomposition of Eq.2 is to establish the relationship between the proposed information-theoretic objective and existing $\beta$-VAE variants, aiming to provide insights regarding the capabilities and limitations of existing methods,  and further motivate the proposed work.
>
> (*) In Table 1, $\lambda_c=0$, $\lambda_a=1$ and $\lambda_b=\beta$ for the original $\beta$-TCVAE is correct. The resaon is that if we decompose the $I(x;z)$ of $\beta$-TCVAE as:
> $$KL(q(z|x)\| p(z))-KL(q(z) \| \prod_{j}q(z_j)-\sum_{j} KL(q(z_j) \| p(z_j)).$$Thus, the regularizer term of $\beta$-TCVAE can be written as: $$\alpha KL(q(z|x)\|p(z)) + (\beta-\alpha) KL(q(z)\|\prod_{j}q(z_j)) + (\gamma-\alpha)\sum_{j=1}^{n} KL(q(z_j) \| p(z_j)).$$ Since $\alpha=\gamma=1$ in $\beta$-TCVAE, therefore when $\lambda_c=0$, the right-hand of Eq.2 is equivalent to $\beta$-TCVAE.

---

> > ### Author Response · Authors · 2021-11-17
> > **Response to Reviewer p3Zz (Part 2/3)**
> >
> > > **Q2:** Do not specify the number of Gate of Encoders (GoE) for different datasets. It is hard to know how many GoE should be used for a new dataset, like CelbeA, that contains 40 latent variables. Also, I am curious about the complexity of the proposed network?
> >
> > **A2:** Thanks for your comment. We first specify the number of Recursive Modules and the corresponding number of Gate-of-Encoders (denoted as Num) for different datasets.
> >
> > |                  | Recursive Module 1 | Num | Recursive Module 2 | Num | Recursive Module 3 | Num |
> > |------------------|--------------------|-----|--------------------|-----|--------------------|-----|
> > | dSprites         | $\surd$            | 4   | $\surd$            | 8   | -                  | -   |
> > | 3DShapes(3DCars) | $\surd$            | 4   | $\surd$            | 10  | -                  | -   |
> > | CelebA           | $\surd$            | 8   | $\surd$            | 8   | $\surd$            | 16  |
> >
> > *Table 1: The details of architecture for different datasets.*
> >
> > Next, we evaluate the **computational complexity** of our model on **3DShapes** and **CelebA**. The evaluation metrics consist of multiply-accumulate operation (MACs), the model parameters (Params), evaluation time and converge time (all models converge to the same reconstruction error as $\beta$-VAE).  As shown in Table 2, the complexity of our model is higher than the vanilla $\beta$-VAE but lower than Factor-VAE. In addition, our model converge faster compared to both baselines.
> >
> > | Dataset  | Method     | MACs(G) | Params(M) | Evaluation time(s) | Converge time(s) |
> > | -------- | ---------- | ------- | --------- | ------------------ | ---------------- |
> > | 3DShapes | RecurD 1   | 3.466   | 3.672     | 0.005124           | 27.1212          |
> > |          | RecurD 2   | 3.469   | 3.694     | 0.005354           | 23.3197          |
> > |          | beta-VAE   | 3.144   | 0.769     | 0.003283           | 32.5251          |
> > |          | Factor-VAE | 3.401   | 4.779     | 0.003389           | 40.6231          |
> > | CelebA   | RecurD 1   | 3.944   | 7.935     | 0.01065            | 29.7421          |
> > |          | RecurD 2   | 3.957   | 8.040     | 0.01087            | 36.6408          |
> > |          | beta-VAE   | 3.145   | 0.769     | 0.01030            | 45.2067          |
> > |          | Factor-VAE | 3.402   | 4.792     | 0.01005            | 48.9939          |
> >
> > *Table 2: Complexity comparison of three models on 3DShapes and CelebA.*
> >
> >
> > >**Q3:** The upper bound and lower bound are confounding. On page 9, the author said: mutual information I(x,z) is the upper bound of KL divergence. In fact, based on the proof in prior work, I(x,z) is the lower bound of KL divergence.
> >
> > **A3:** Thanks for pointing it out. You are correct. It should be lower bound. We have fixed the typo in Page.9 (Result Discussion).
> >
> >
> > > **Q4:** The Markov Chain in Eq.7 is incorrect. Based on my understanding, the next state $X_{t+1}$ of Markov chain is only related to the current state $X_t$. Hence the joint probability of $p(a,b,c) = p(a)p(b|a)p(c|a)$ rather than $p(a)p(b|a)p(c|b)$. This is because $c$ is conditionally independent on b as you mentioned, $b\rightarrow a \rightarrow c$.
> >
> > **A4:** Thanks for pointing it out. It should be $p(a)p(b|a)p(c|a)$ to be consistent with discussion about Markov Chain Property. We have fixed the typo in Eq.7 (Page.11 Appendix A).

---

> > > ### Author Response · Authors · 2021-11-17
> > > **Response to Reviewer p3Zz (Part 3/3)**
> > >
> > > > **Q5:** Missing baselines in recent years. The author only discussed and compare the proposed method with the baselines before the year of 2019. There are some recent works on improving the disentanglement and reconstruction error. For instance, ControlVAE dynamically tunes the weight $\beta$ on the KL term to achieve a good trade-off between disentanglement and reconstruction error.
> > >
> > > **A5:** Thanks for your comment. Following your suggestion, we add ControlVAE[1] and NVAE[2] as new baselines on dSprites and 3DShapes. As shown in Table 3, RecurD outperforms ControlVAE on disentanglement quality and reconstruction error. NVAE can achieve comparable reconstruction error compared to RecurD. However, its disentanglement qualities are not as good as RecurD, because RecurD additionally regularizes the inter-layer information sharing and alleviate the information redundancy of multiple layers. This experiment demonstrates the superiority of the proposed RecurD method in disentanglement compared to existing hierarchical VAEs.
> > >
> > > |          | ControlVAE |                  |  NVAE  |                  | RecurD |                  |
> > > | :------: | :--------: | ---------------- | :----: | ---------------- | :----: | ---------------- |
> > > |          |    MIG     | Reconst. Error |  MIG   | Reconst. Error |  MIG   | Reconst. Error |
> > > | dSprites |   0.2455   | 0.0102           | 0.0043 | 0.0041           | 0.2707 | 0.0047           |
> > > | 3DShapes |   0.2630   | 0.0357           | 0.0081 | 0.0078           | 0.3105 | 0.0083           |
> > >
> > > *Table 3: Comparison with ControlVAE and NVAE on dSprites and 3DShapes.*
> > >
> > > Note that Robst[3] is a specific method since the training process requires pairs of images depicting the same object appearance, and MS-VAE[4] has no public code, so we do not add them as compared methods.
> > >
> > > > **Q6:** Did not conduct experiments on complex datasets. The authors should do experiments on 3D chairs or CelebA to demonstrate the good performance of the proposed method.
> > >
> > > **A6:** Thanks for your comment. Following your suggestion, we conduct quantitative experiments on 3DCars, and qualitative experiments on CelebA (the cropped version used by previous works). We present quantitative results on 3DCars as below (Table 4), which demonstrate the advantage of our method on disentanglement performance over the three baselines ($\beta$-VAE, NVAE[2], and ControlVAE[1]) on complex datasets 3DCars and CelebA. The qualitative results on cropped CelebA are included in Appendix D.3. On CelebA, our model can disentangle eight factors, including old, azimuth, beard, bangs, face width, brightness, glasses and skin color. These experiments demonstrate the capability of our method on complex datasets 3DCars and CelebA.
> > >
> > > |                | RecurD | $\beta$-VAE | NVAE   | ControlVAE |
> > > | -------------- | ------ | ----------- | ------ | ---------- |
> > > | MIG            | 0.1762 | 0.1015      | 0.0034 | 0.1583     |
> > > | Reconst. Error | 0.0132 | 0.0376      | 0.0118 | 0.0257     |
> > >
> > > *Table 4: Quantitative comparison on 3DCars.*
> > >
> > > > **Q7:** The result in Fig 5 does not perform well. We can observe that for orientation and scale factors, they are slightly entangled. Besides, the reconstruction quality is not as good as ControlVAE and FactorVAE in the paper. In particular, ControlVAE and FactorVAE can disentangle both 5 latent factors which are better than those in this work.
> > >
> > > **A7:** Thanks for your comment. However, it is unclear to us which part of the result in Fig. 5 does not perform well. We have enlarged the figure and included it in the Appendix D.1. It would be great if you can help elaborate more the problem. In addition, the reconstruction quality can be quantitatively measured by the reconstruction error, and the results shown in Table 5 demonstrate our method achieves lower error than ControlVAE and FactorVAE.
> > >
> > > |          | ControlVAE     | Factor-VAE     | RecurD         |
> > > |:--------:| -------------- | -------------- | -------------- |
> > > |          | Reconst. Error | Reconst. Error | Reconst. Error |
> > > | dSprites | 0.0102         | 0.0228         | 0.0047         |
> > > | 3DShapes | 0.0357         | 0.0800         | 0.0083         |
> > >
> > > *Table 5: Quantitative comparison of reconstruction quality.*
> > >
> > > > **Q8:** There are some typos in this paper, please proofread this manuscript. For instance, priopr work $\rightarrow$ prior work on Page 5.
> > >
> > > **A8:** Thanks for pointing them out. We have proofread the manuscript and fixed the typos.

---

> > > > ### Author Response · Authors · 2021-11-17
> > > > **Response to Reviewer p3Zz (Reference)**
> > > >
> > > > ---
> > > > **Reference:**
> > > >
> > > > [1] [Controlvae: Controllable variational autoencoder.](https://arxiv.org/abs/2004.05988) Shao, H., Yao, S., Sun, D., Zhang, A., Liu, S., Liu, D., ... & Abdelzaher, T. International Conference on Machine Learning. PMLR, 2020.
> > > >
> > > > [2] [VAE with a VampPrior.](https://arxiv.org/abs/1705.07120) Tomczak, Jakub, and Max Welling. International Conference on Artificial Intelligence and Statistics. PMLR, 2018.
> > > >
> > > > [3] [Unsupervised Robust Disentangling of Latent Characteristics for Image Synthesis.](http://openaccess.thecvf.com/content_ICCV_2019/html/Esser_Unsupervised_Robust_Disentangling_of_Latent_Characteristics_for_Image_Synthesis_ICCV_2019_paper.html) Esser, Patrick, Johannes Haux, and Bjorn Ommer. ICCV 2019.
> > > >
> > > > [4] [Improving the Reconstruction of Disentangled Representation Learners via Multi-Stage Modelling.](https://arxiv.org/abs/2010.13187) Srivastava, Akash, Yamini Bansal, Yukun Ding, Cole Hurwitz, Kai Xu, Bernhard Egger, Prasanna Sattigeri, Josh Tenenbaum, David D. Cox, and Dan Gutfreund. arXiv preprint arXiv:2010.13187 (2020).
> > > >
> > > > [5] [Variational Discriminator Bottleneck: Improving Imitation Learning, Inverse RL, and GANs by Constraining Information Flow.](https://arxiv.org/abs/1810.00821) Xue Bin Peng, Angjoo Kanazawa, Sam Toyer, Pieter Abbeel, Sergey Levine. ICLR 2019.

---

> > > ### Comment · Reviewer_p3Zz · 2021-11-23
> > > **About Computational Complexity**
> > >
> > > First, thanks a lot for fixing the typos.
> > > In addition, what does converge time mean? Is it a training time? I am a bit confused.

---

> > > > ### Author Response · Authors · 2021-11-23
> > > > **Response to Reviewer p3Zz**
> > > >
> > > > Yes,  the converge time means the training time, i.e., the time cost when each model converges to the same reconstruction error as $\beta$-VAE.

---

> > > > > ### Comment · Reviewer_p3Zz · 2021-11-23
> > > > > **About Training Time**
> > > > >
> > > > > Do you mean it only takes 50 seconds to train the beta-VAE? I am a bit confused. I am wondering if you could clarify the details of measuring the converge time?

---

> > > > > > ### Author Response · Authors · 2021-11-23
> > > > > > **Response to Reviewer p3Zz**
> > > > > >
> > > > > > The experimental setting is: the learning rate is set as 5e-3 and the batch size is set as 64. The converge time is measured based on the reconstruction loss, which is a common loss term for all models, so it is used for a fair comparison. Note that the reconstruction loss term converges very quickly. By the time of its convergence,  other model-specific regularization loss terms have not converged yet, so it does not mean it only takes 50 seconds to train the Beta-VAE.

---

> > > > > > > ### Comment · Reviewer_p3Zz · 2021-11-23
> > > > > > > **Thanks for Response**
> > > > > > >
> > > > > > > My concern is about computational complexity. I am wondering if you could report the training time for each model?

---

> > > > > > > > ### Author Response · Authors · 2021-11-23
> > > > > > > > **Response for the training time**
> > > > > > > >
> > > > > > > > We randomly select 30,000 samples from 3DShapes to quickly evaluate the training time. The learning rate is set as 5e-3 and batch size is set as 64. The training times of Beta-VAE, Factor-VAE, and our method are 160s, 305s, and 286s, respectively.

---

> > ### Comment · Reviewer_p3Zz · 2021-11-23
> > **Thanks for the Clarification**
> >
> > Thanks a lot for the clarification and explanation. I understand it now.

---

### Official Review · Reviewer_1v1o · 2021-10-27

**Correctness:** 4
**Technical Novelty And Significance:** 3
**Empirical Novelty And Significance:** 3
**Recommendation:** 6
**Confidence:** 4

**Main Review:**

Strengths
+ The paper presents an approach that is theoretically justified and implemented in a convincing manner.
+ The results appear convincing and beating relevant baselines (though with the disclaimer that mostly these baselines seem rather old by now; some of the relevant comparisons from the last 2 years might be missing, but I cannot name any).
+ Empirical evaluations are sufficient (though only barely), except for the ablation (*):

Weaknesses
- I suspect that including layer-wise disentanglement has occurred to many people before, but it has not been attempted due to the computational burden. (That said, I have not seen anyone actually try it.)
- It is unclear whether this approach *solves* the computational burden and scales to more complex datasets and larger image resolution
- I am not convinced that the loss formulation (Eq. 4) is that significant wrt. InfoVAE, for example. What would the authors say against the criticism that Eq. 4 is basically just reshuffling the InfoVAE loss? (I'm not saying it is, but it would be helpful if the authors could shoot down this potential concern.)
- (*) It is unclear to what extent does the performance originate from the loss (Equation 4), and to what extent does it come from the switch-based architecture. Could the authors clarify this? It would seem that one could implement Eq. 4 loss without the switch architecture?

**Summary Of The Paper:**

The paper presents a VAE variant that disentangles the features of the inference network at every layer, with disentanglement defined in terms of mutual information between features. The approach is implemented as "recursive disentanglement network" based on a switch network (aka Mixture-of-Experts gate, introduced in Shazeer 2017 and used in the switching transformers). The results in dSprites and 3DShapes dataset suggest this variant performs better than well-known disentanglement VAE networks (from few years back) in dSprites and well in 3DShapes (though not the best in all measures). In terms of VAE loss, the approach is presented as a generalization of various other disentangling VAEs.

**Summary Of The Review:**

The paper appears to simultaneously give a clear slightly novel generalization to disentanglement losses of prior VAE variants, and to provide an architectural approach to implement their approach.

I have concerns about the scalability, and I am suspicious about whether such a heavy-handed disentanglement can be maintained in larger models.

It also was not clear to me whether the results originate from the loss or the architecture advances (which build on existing switching architectures).

However, either way, I think this is a novel approach and potentially a significant addition to the VAE disentanglement research, and I lean towards acceptance in this case, though I urge the authors to address my questions.

---

> ### Author Response · Authors · 2021-11-17
> **Response to Reviewer 1v1o (Part 1/3)**
>
> > **Q1:** I suspect that including layer-wise disentanglement has occurred to many people before, but it has not been attempted due to the computational burden. (That said, I have not seen anyone actually try it.)
>
> **A1:** Thanks for your comment. Indeed, recent work on hierarchical VAEs introduces layer-wise disentanglement regularization. We have included related hierarchical models and clarified the differences in the revised manuscript (Section X), also quoted below.
>
> Prior works on hierarchical VAEs, e.g., VampPrior[1], Ladder VAEs[2], and NVAE[3], introduce layer-wise disentanglement regularization to learn conditioning structures across multi-layer latent variables.
>
> In existing hierarchical model structures, e.g., the cross-layer residual connection structure in NVAE[3] and VampPrior[1],  inter-layer regularization is less of a focus. The latent variables of a preceding layer serve as shared inputs to the next layer, which introduces information redundancy and hence impairs representation disentanglement. In contrast, the proposed compositional objective optimizes the statistical independence of inter-layer and intra-layer latent variables simultaneously, thereby minimizing the information redundancy of inter-layer information sharing and improving disentanglement quality.
>
> We have conducted the following study to compare the proposed RecurD with NVAE[3] and VampPrior[1] on 3DShapes. This study has been included in the Appendix D.1.
>
> As shown in Table 1, NVAE and VampPrior can achieve comparable reconstruction error compared to RecurD. However, their disentanglement qualities are not as good as RecurD, because RecurD additionally regularizes the inter-layer information sharing and alleviate the information redundancy of multiple layers. This experiment demonstrates the superiority of the proposed RecurD method in disentanglement compared to existing hierarchical VAEs.
>
> |                | RecurD | $\beta$-VAE | NVAE   |VampPrior|
> | -------------- | ------ | ----------- | ------ | --------|
> | MIG            | 0.3105 | 0.2519      | 0.0081 | 0.1423 |
> | Reconst. Error |0.0083  | 0.0216      | 0.0078 | 0.0089 |
> *Table 1: Comparison with hierarchical VAEs on 3DShapes.*

---

> > ### Author Response · Authors · 2021-11-17
> > **Response to Reviewer 1v1o (Part 2/3)**
> >
> > > **Q2:** It is unclear whether this approach solves the computational burden and scales to more complex datasets and larger image resolution.
> >
> > **A2:** Thanks for your comment. First, we evaluate the **computational complexity** of our model on **3DShapes** and **CelebA**.The evaluation metrics consist of multiply-accumulate operation (MACs), the model parameters (Params), evaluation time and converge time (all models converge to the same reconstruction error as $\beta$-VAE).
> >
> > As shown in Table 2, the complexity of our model is higher than the vanilla $\beta$-VAE but lower than Factor-VAE. In addition, our model converges faster compared to both baselines.
> >
> > | Dataset  | Method     | MACs(G) | Params(M) | Evaluation time(s) | Converge time(s) |
> > |----------|------------|---------|-----------|--------------------|------------------|
> > | 3DShapes | RecurD~1   | 3.466   | 3.672     | 0.005124           | 27.1212          |
> > |          | RecurD~2   | 3.469   | 3.694     | 0.005354           | 23.3197          |
> > |          | beta-VAE   | 3.144   | 0.769     | 0.003283           | 32.5251          |
> > |          | Factor-VAE | 3.401   | 4.779     | 0.003389           | 40.6231          |
> > | CelebA   | RecurD~1   | 3.944   | 7.935     | 0.01065            | 29.7421          |
> > |          | RecurD~2   | 3.957   | 8.040     | 0.01087            | 36.6408          |
> > |          | beta-VAE   | 3.145   | 0.769     | 0.01030            | 45.2067          |
> > |          | Factor-VAE | 3.402   | 4.792     | 0.01005            | 48.9939          |
> >
> > *Table 2: Complexity comparison of three models on 3DShapes and CelebA.*
> >
> > To evaluate the performance of our model on complex datasets, we conduct experiments on 3DCars and CelebA (the cropped version used in previous works). The qualitative results on CelebA are presented in Appendix D.3. On CelebA, our model can disentangle eight factors, including old, azimuth, beard, bangs, face width, brightness, glasses and skin color.
> >
> > In addition, we compare the quantitative results on 3DCars with two recent works, i.e., NVAE[3] and VAE[4]. As shown in Table 3, the results demonstrate the performance advantage of the proposed recursive disentanglement network on complex datasets. This study has been included in the Appendix D.1.
> >
> > |                | RecurD | $\beta$-VAE |  NVAE  | ControlVAE |
> > |:--------------:|:------:|:-----------:|:------:|:----------:|
> > |       MIG      | 0.1762 |    0.1015   | 0.0034 |   0.1583   |
> > | Reconst. Error | 0.0132 |    0.0376   | 0.0118 |   0.0257   |
> > *Table 3: Quantitative comparison on 3DCars.*
> >
> > Furthermore, we suggest the following two approaches which may further alleviate the computation burden.
> >
> > 1) GoE hyperparameter adjustment, e.g., the number of encoders and hyperparameter k in the switch strategy. In addition, the model can be described in mixed-precision using the NVIDIA APEX library to reduce the memory requirement.
> > 2) Consider other architectures with potentially lower computational cost,  e.g., fixed assignment, multiple parallel branches with linear and softmax and multi-head attention mechanism.
> >
> > > **Q3:** I am not convinced that the loss formulation (Eq. 4) is that significant wrt. InfoVAE, for example. What would the authors say against the criticism that Eq. 4 is basically just reshuffling the InfoVAE loss? (I'm not saying it is, but it would be helpful if the authors could shoot down this potential concern.)
> >
> > **A3:** Thanks for your comment. InfoVAE[5] aims to learn an amortized inference distribution $q_\phi(z|x)$ to approximate the true posterior $p_\theta(z|x)$. Its objective is used to encourage high correlation between the final latent variables and data.
> > In contrast, our objective is to encourage disentanglement across the whole layer-wise compositional feature space, in addition to the final latent variables.
> > We conduct the following experiment and compare InfoVAE with RecurD on 3DShape. As shown in Table 4, RecurD achieves significantly lower reconstruction error as well as higher disentanglement score compared to InfoVAE.
> >
> > |                | InfoVAE | RecurD |
> > |:--------------:|:-------:|:------:|
> > |       MIG      |  0.2108 | 0.3105 |
> > | Reconst. Error |  0.0534 | 0.0083 |
> >
> > *Table 4: Comparison with InfoVAE on 3DShapes.*

---

> > > ### Author Response · Authors · 2021-11-17
> > > **Response to Reviewer 1v1o (Part 3/3)**
> > >
> > > > **Q4:** (*) It is unclear to what extent does the performance originate from the loss (Equation 4), and to what extent does it come from the switch-based architecture. Could the authors clarify this? It would seem that one could implement Eq. 4 loss without the switch architecture?
> > >
> > > **A4:** Thanks for your comment. Following your suggestion, we have conducted two additional ablation experiments on 3DShapes to evaluate the impact of the proposed architecture and compositional objective. The results are presented in the Appendix D.2.
> > >
> > > The first experiment evaluates the impact of the proposed architecture.  We implement three variants of RecurD with the same Eq.4 loss and varying architecture of Gate-of-Encoders (GoE), including **Linear**: GoE is implemented as a linear with softmax; **Fix**: GoE is implemented as fix assignment, i.e., we split $m_l$ in $d+1$ equal slices; and **Att**: GoE is implemented as a multi-head attention layer (the head is fixed as $8$).
> > > The results of MIG are shown as below. As shown in Table 5, all GoE architectures improves disentanglement quality compared to $\beta$-VAE, and the proposed GoE offers the best performance.
> > >
> > > |                | Linear |   Fix  |   Att  | RecurD | $\beta$-VAE |
> > > |:--------------:|:------:|:------:|:------:|:------:|:-----------:|
> > > |       MIG      | 0.2539 | 0.2617 | 0.2604 | 0.3105 |    0.2519   |
> > > | Reconst. Error | 0.0176 | 0.0116 | 0.0095 | 0.0083 |    0.0216   |
> > >
> > > *Table 5: Ablation study on the GoE architecture.*
> > >
> > > The second experiment evaluates the impact of compositional learning objective. We compare performance between the same RecurD with 3 Recursive Modules but on the standard learning objective (Eq.1) and the compositional learning objective (Eq.4), respectively. The results shown in Table 6 demonstrate that the same architecture RecurD performs better when it is implemented on the compositional learning objective.
> > >
> > > |                | RecurD (with Eq.1) | RecurD (with Eq.4) |
> > > |:--------------:|:-------------:|:-------------:|
> > > |       MIG      |     0.2622    |     0.3105    |
> > > | Reconst. Error |     0.0143    |     0.0083    |
> > >
> > > *Table 6: Performance comparison of the standard and compositional learning objective.*
> > >
> > > In summary, the two sets of results shown above indicate the benefit of propagating disentanglement across the compositional feature space and the switch-based architecture.
> > >
> > > ---
> > > **Reference:**
> > >
> > > [1] [VAE with a VampPrior.](https://arxiv.org/abs/1705.07120) Tomczak, Jakub, and Max Welling. International Conference on Artificial Intelligence and Statistics. PMLR, 2018.
> > >
> > > [2] [Ladder variational autoencoders.](https://arxiv.org/abs/1602.02282) Sønderby, Casper Kaae, et al. Advances in neural information processing systems 29 (2016): 3738-3746.
> > >
> > > [3] [Nvae: A deep hierarchical variational autoencoder.](https://arxiv.org/abs/2007.03898) Vahdat, Arash, and Jan Kautz. arXiv preprint arXiv:2007.03898 (2020).
> > >
> > > [4] [Controlvae: Controllable variational autoencoder.](https://arxiv.org/abs/2004.05988) Shao, H., Yao, S., Sun, D., Zhang, A., Liu, S., Liu, D., ... & Abdelzaher, T. International Conference on Machine Learning. PMLR, 2020.
> > >
> > > [5] [Infovae: Information maximizing variational autoencoders.](https://arxiv.org/abs/1706.02262) Zhao, Shengjia, Jiaming Song, and Stefano Ermon. arXiv preprint arXiv:1706.02262 (2017).

---

> > > ### Comment · Reviewer_1v1o · 2021-11-22
> > > **Response**
> > >
> > > Thank you for the detailed responses. While considerable work has been done to address the questions of the reviewers, I remain divided about the paper, however.
> > >
> > > In short, I remain concerned about how the model scales up. While the performance appears fine, the new tables provided also show that the number of parameters on e.g. CelebA is 10 times the number of Beta-VAE. Therefore, is the comparison really fair to Beta-VAE?
> > >
> > > The added complexity is especially worrisome, given the unimpressive sample quality of CelebA faces (fig20-27). Despite the aggressive cropping, they are often very fuzzy, to the point of not often really demonstrating the disentanglement of the features in question. Now, whether they are fuzzier than some of the prior VAEs is a moot point, since the level of fuzziness in some of those papers (e.g. FactorVAE), too, is pretty bad on today's standards. It is simply that the amount of information conveyed in a blurry image is relatively minor, and hence does not provide the intended empirical support to the claims, when the features boil down to really just a Gaussian mass of color.
> > >
> > > Sorry for my late reply, but could the authors provide a final comment on this?

---

> > > > ### Author Response · Authors · 2021-11-23
> > > > **Response to Reviewer 1v1o**
> > > >
> > > >
> > > >
> > > > Thanks for your comment. In terms of parameter efficiency, as shown in the table below, the recursive disentanglement network itself only contains 0.826 million parameters (RecurD 2 w/o MINEs), which are comparable to Beta-VAE. In fact, most of the parameters of RecurD 2 are contributed by the MINE estimator. Specifically, in the original model implementation (RecurD 2/specific MINEs), each pair of outputs of encoders is equipped with a specific MINE model, which contributes 7.214 million parameters in total. Following your comment, we have further optimized the design by using shared MINE within the same feature category, which can effectively reduce the total number of parameters down to 1.325 million (RecurD 2/shared MINEs). We have included this table in Appendix D.4.
> > > >
> > > > The fuzziness of traversal samples comes from the limited information capacity of the model. Following your comment, we tentatively increase the dimensionality of latent variables of RecurD (from 16 to 32) by doubling the output of the last layer of each encoder from a single latent variable to a 2-dimensional variable.  Using this new model setup, we present two traversal examples in Figure 21 in Appendix D.3. As we can see, the generated samples become much clearer. Due to limited time for rebuttals, we currently only include two examples, and we will include more results in the final version.
> > > >
> > > > To further illustrate the efficacy of recursive disentanglement, we conduct another study by further reducing the dimensionality of the latent variables of Beta-VAE and RecurD (from 16 to 8) and comparing the traversal samples generated by the two models. The rationale behind this experimental setup is that a method with better disentanglement capability can more effectively eliminate information redundancy, and pass more non-redundant information across the information bottleneck. As shown in Figure 20 in Appendix D.3, as the dimensionality of the latent variable decreases, Beta-VAE can only disentangle four factors while RecurD can disentangle six factors with two additional factors.
> > > >
> > > > In addition, we also measure the reconstruction error of AE, which may be regarded as an upper limit of Beta-VAE’s performance in terms of reconstruction error. From the table below, we can see that Recurd 2 provides reconstruction error comparable to AE, and both are significantly better than Beta-VAE.
> > > >
> > > >
> > > >
> > > > |                           | MACs(G) | Params(M) | Reconst. Error |
> > > > | ------------------------- | ------- | --------- | -------------- |
> > > > | RecurD 2/specific MINEs | 3.957   | 8.040     | 0.01749        |
> > > > | RecurD 2 w/o MINEs      | 2.960   | 0.826     | /              |
> > > > | RecurD 2/shared MINEs   | 3.654   | 1.325     | 0.01769        |
> > > > | Beta-VAE                  | 3.145   | 0.769     | 0.03609        |
> > > > | Factor-VAE                | 3.402   | 4.792     | 0.05681        |
> > > > | AE                        | 3.145   | 0.765     | 0.01657        |
> > > >
> > > > *Table 7: Complexity comparsion.*

---

> > > > > ### Comment · Reviewer_1v1o · 2021-11-23
> > > > > **Thanks for the final clarifications**
> > > > >
> > > > > Thank you for the additional information and experiments. This addresses my concerns to some extent. While the latent dimensionality indeed makes a difference in the image quality, which is not surprising (though important to check), it is now treated in isolation. Hence, still not crystal clear to me how the model will scale in other respects (parameters, training time, inference time) with respect to the increase in latent dimensionality. I realize my prior comment was late and it is of course not reasonable for me to ask the authors for more information at this final hour of the discussion period.
> > > > >
> > > > > As a general point, in terms of the questions of scaling, complexity and sample quality, I think it would have made the discussion more straight-forward if the authors would have started out with a larger latent dimensionality to begin with. I just want to emphasize that given the limited time to digest a paper, it is difficult to grok the subtleties of the multi-dimensional hyper-parameter space, and it is easy to overlook its significance. Thus, with all due respect to the authors, whenever one observes that a model has been only measured in terms of a rather small architecture, it raises the question of whether it is bad overall practical scalability that forces the authors of the paper to only use the small architecture, whether this is the case or not.
> > > > >
> > > > > Again, thank you for the additional efforts.

---

> > > > > > ### Author Response · Authors · 2021-11-23
> > > > > > **Response to Reviewer 1v1o**
> > > > > >
> > > > > > Thank you for your understanding.
> > > > > >
> > > > > > The motivation of this work is based on the latest developments in Beta-VAE-based disentangled representation learning. Therefore, for comparison purposes, the experimental settings (such as model architecture and datasets) have been aligned with the settings used by existing Beta-VAE and its variants.
> > > > > >
> > > > > > We agree with the reviewer that large-scale architecture will help evaluate the potential of the proposed work. As part of our future work, we are working in this direction.

---

### Official Review · Reviewer_VLG5 · 2021-11-02

**Correctness:** 4
**Technical Novelty And Significance:** 3
**Empirical Novelty And Significance:** 3
**Recommendation:** 6
**Confidence:** 3

**Main Review:**

Overall, the idea is technically sound and the results look promising. I have some questions and suggestions and hope the authors could clarify them during the rebuttal.

* Section 2.1: what is the meaning of defining Markov Chain as \hat{x} -> x -> z, given that the generative process is actually x -> z -> \hat{x}? I looked at the cited work (Achille & Soatto, 2018), and they seem to discuss a different setting, where they have a dataset of data points x and associated labels y, and in that case, y -> x -> z makes sense to me.

* Definition 5: Similarly, what is the practical meaning of m_j^{l+1} -> x -> (m_i^l,m_j^l)?

* Figure 6 left: do you use compositional objective (Section 2.2) and recursive disentanglement network (Section 3) in this experiment? If so, lambda_2=0 is not equivalent to beta-VAE as the vanilla beta-VAE does not have the compositional objective.

* Figure 2: the 'w_j^l' in green should be 'w_{d_{l+1}}^l'?

* I would suggest adding GAN-based approaches into the comparison tables. This would be very helpful for readers who want to pick techniques for their downstream applications.

* The decomposition and discussion of the losses around Eq. 2 and Table 1 have been partially discussed in prior work (e.g., https://arxiv.org/abs/1706.02262), and I believe it is not your key contribution. I would suggest highlighting this fact better as it sounds like these are your discoveries from the current writing.

**Summary Of The Paper:**

The paper proposes a new approach for learning disentangled variational autoencoders. In addition to pushing the sufficiency, minimal sufficiency, and disentanglement of the latent representation, the paper proposes to also regularize those on earlier features in the network. Experiments demonstrate promising results.

**Summary Of The Review:**

In summary, the idea is interesting and the results look promising. I hope the readers could clarify these questions and I will adjust the score accordingly.

---

> ### Author Response · Authors · 2021-11-17
> **Response to Reviewer VLG5**
>
> > **Q1:** Section 2.1: what is the meaning of defining Markov Chain as $\hat{x} \rightarrow x \rightarrow z$, given that the generative process is actually $x \rightarrow z \rightarrow \hat{x}$? I looked at the cited work (Achille, Soatto, 2018), and they seem to discuss a different setting, where they have a dataset of data points x and associated labels y, and in that case, $y\rightarrow x\rightarrow z$ makes sense to me.
>
> **A1:** Thanks for your comment. First, $\hat{x}$ in our work and $y$ in the cited work represent the same thing. For unsupervised image reconstruction, given a test random variable $x$, both the cited work and our work aim to infer the random variable, $y$ (the cited work) or $\hat{x}$ (our work). In our work, to infer the random variable $\hat{x}$, $z$ is the latent representation give data $x$, that is, the distribution of $z$ depends on $\hat{x}$ only through $x$, as expressed by Markov Chain as $\hat{x} \rightarrow x \rightarrow z$, please also see the work of [1]. We then formulate sufficiency and minimal sufficiency using the Data Processing Inequality property of Markov Chain.
>
>
> > **Q2:** The meaning of $m_j^{l+1} \rightarrow x \rightarrow (m_i^l,m_j^l)$
>
> **A2:** Thanks for your comment. $m^l$ and $m^{l+1}$ denote the input and output latent representation of the $l$-th layer Recursive Module, respectively. The $m_j^{l+1} \rightarrow x \rightarrow (m_i^l,m_j^l)$ Markov Chain aims to infer $m_j^{l+1}$, the output latent representation of the $l$-th layer Recursive Module, with $m^l$ ($m_i^l$ and $m_j^l$) as the input latent representation of the $l$-th	layer Recursive	Module given data $x$, i.e., the distribution of $m^{l}$ depends on $m^{l+1}$ only through $x$. In this work, the compositional minimal sufficiency is defined based on the aforementioned Markov Chain.
>
> > **Q3:** Figure 6 left: do you use compositional objective (Section 2.2) and recursive disentanglement network (Section 3) in this experiment? If so, $\lambda_2=0$ is not equivalent to beta-VAE as the vanilla beta-VAE does not have the compositional objective.
>
> **A3:** Yes, the proposed compositional objective (Section 2.2) and recursive disentanglement network (Section 3) are used in the experiment shown in Figure 6. Yes, you are correct. RecurD with $\lambda_2=0$ reduces to $\beta$-VAE with the proposed compositional architecture. We have fixed Figure 6 caption in the manuscript accordingly.
>
> > **Q4:** Figure 2: the $w_j^l$ in green should be $w_{d_{l+1}}^l$？
>
> **A4:** Thanks for pointing it out. We have fixed the typo.
>
> > **Q5:** I would suggest adding GAN-based approaches into the comparison tables. This would be very helpful for readers who want to pick techniques for their downstream applications.
>
> **A5:** Thanks for your comment. Following your suggestion, we compare the disentanglement score with InfoGAN [2]. Table 1 presents MIG on 3Dshape, which confirms the performance advantage of the proposed method compared against GAN-based approaches. Due to space limitation, the detailed results are included in the Appendix D.1.
>
> |      | Info-GAN | RecurD |
> | ---- | -------- | ------ |
> | MIG  | 0.1874   | 0.3105 |
>
> *Table 1: Comparison with Info-GAN.*
>
> > **Q6:** The decomposition and discussion of the losses around Eq. 2 and Table 1 have been partially discussed in prior work (e.g., https://arxiv.org/abs/1706.02262), and I believe it is not your key contribution. I would suggest highlighting this fact better as it sounds like these are your discoveries from the current writing.
>
> **A6:** Thanks for your comment. We agree that some prior work has touched on Eq. 2. We have included the related works [3] and [4] in Section 2.1.
>
> ---
> **Reference:**
>
> [1] [Information dropout: Learning optimal representations through noisy computation.](https://arxiv.org/abs/1611.01353) Achille, Alessandro, and Stefano Soatto. IEEE transactions on pattern analysis and machine intelligence 40.12 (2018): 2897-2905.
>
> [2] [Infogan: Interpretable representation learning by information maximizing generative adversarial nets.](https://arxiv.org/abs/1606.03657) Chen, Xi, et al. Proceedings of the 30th International Conference on Neural Information Processing Systems. 2016.
>
> [3] [Infovae: Information maximizing variational autoencoders.](https://arxiv.org/abs/1706.02262) Zhao, Shengjia, Jiaming Song, and Stefano Ermon. arXiv preprint arXiv:1706.02262 (2017).
>
> [4] [PRI-VAE: principle-of-Relevant-Information variational autoencoders.](https://arxiv.org/abs/2007.06503) Li, Yanjun, et al. arXiv preprint arXiv:2007.06503 (2020).

---

> > ### Comment · Reviewer_VLG5 · 2021-11-24
> > **Thank you**
> >
> > Thank you for the explanations, which cleared my questions.

---

### Official Review · Reviewer_yTV3 · 2021-11-02

**Correctness:** 4
**Technical Novelty And Significance:** 4
**Empirical Novelty And Significance:** 3
**Recommendation:** 6
**Confidence:** 3

**Main Review:**

Overall, I am pretty happy with the paper. It's mostly well written and organised.

1. Positives

* Session 2 (Compositional disentanglement learning) is well organised and sets up the scene well for the method in session 3.
* Good level of implementation detail is available, such as architectures, estimator used, etc. The experiments are well conducted and common mistakes were avoided AFAICT.
* Use of standard datasets and metrics well stablished in the field.


2. For improvement:

* There has been some progress in the use of hierarchical VAEs, which can be interpreted as applying disentanglement regularisation to other layers and making it compositional in a similar fashion to this work, e.g. NVAE (A Vahdat, J Kautz 2020).
* I would be a bit more careful with the tone on claims about the requirement of compositionality for disentanglement. Figure 1 is only an evidence in a toy example, not an actual demonstration. So statements as " fig 1 shows when ... is not effectively disentangled." (session 1) and "To achieve better disentanglement between ... their input feature sets .. are expected to be disentangled as demonstrated in our case study" (session 2.2) could be watered down a little.
* It's unclear if there are benefits from using MINE and the architecture in some of the experiments. For example, if I understood correctly, the beta-VAE objective yields better metrics on figure 6 than in figure 3.
* The paper left me wondering what are the disentanglement metrics on the preceding layers, looking into MIG/SAP/DCI-D on m^L-1 and m^L-2 seems like a straightforward analysis that should be in the appendix or even in the main paper.
* Some ablations on the architecture itself also seem to be missing (ablations on the loss are good). My intuition is that Gate of Mixture-of-Experts fits quite well with the disentanglement that we want, because of the top . For example:
  * just learning a linear+softmax instead of the Router
  * no routing at all, just some fixed assignment: e.g. split m_l in d+1 equal slices and pass through the encoders.
  * using a transformer instead of the Recursive Modules, perhaps over fixed slices as well.

**Summary Of The Paper:**

This paper formulates the disentanglement problem from an information theoretic perspective, but focusing on an objective that encourages a compositional disentangled feature space on the layers that precede the final latents.

With objective, the authors describe a new method using Gate of Mixture-of-Experts to implement the compositional disentangled reconstruction objective. Some of the terms require mutual information estimation, for which they use MINE estimators.

They run experiments across dSprites and 3DShapes and look into reconstruction error and different disentanglement metrics, observing that they method outperform existing beta-VAE-like baselines, without any compositional incentives. They also analyse the loss components with different architectures and observe that degrees of compositionally in the architecture yields better disentanglement.

Finally, they look into some ablations of the regularisation pressure and into data efficiency in downstream tasks.



**Summary Of The Review:**

My main concern is the discussion of related hierarchical models missing from related work and the emphasis on this being the only work to apply some disentanglement pressure outside the main z latents. This should be an easy fix for this paper.

The compositional objective is interesting and novel and the implementation method is clean. The experiments were well conducted and the well analysed. Overall, I am confident that the authors will be able to address the main issue above and that this paper will award acceptance in this venue.

---

> ### Author Response · Authors · 2021-11-17
> **Response to Reviewer yTV3 (Part 1/2)**
>
> > **Main Concern:** My main concern is the discussion of related hierarchical models missing from related work and the emphasis on this being the only work to apply some disentanglement pressure outside the main z latents.
> >
> > **Q1:** There has been some progress in the use of hierarchical VAEs, which can be interpreted as applying disentanglement regularization to other layers and making it compositional in a similar fashion to this work, e.g. NVAE (AVahdat, JKautz 2020).
>
> **AM/A1:** Thanks for your comment. Indeed, recent works on hierarchical VAEs introduce layer-wise disentanglement regularization. We have included related hierarchical models and clarified the differences in the revised manuscript (Section 4), also quoted below.
>
> Prior works on hierarchical VAEs, e.g., VampPrior[1], Ladder VAEs[2], and NVAE[3], introduce layer-wise disentanglement regularization to learn conditioning structures across multi-layer latent variables. Specially, VampPrior learns a complex prior to reduce over-regularization for disentanglement learning[1]. Ladder VAEs improve generative performance by introducing skip connections during the stochastic sampling process[2]. NVAE optimizes high-quality image generation via global correlation capturing across multi-layer latent variables[3].
>
> In existing hierarchical model structures, e.g., the cross-layer residual connection structure in NVAE[3] and VampPrior[1], inter-layer regularization is less of a focus. The latent variables of a preceding layer serve as shared inputs to the next layer, which introduces information redundancy and hence impairs representation disentanglement. In contrast, the proposed compositional objective optimizes the statistical independence of inter-layer and intra-layer latent variables simultaneously, thereby minimizing the information redundancy of inter-layer information sharing and improving disentanglement quality.
>
> We have conducted the following study to compare the proposed RecurD with NVAE[3] and VampPrior[1] on 3DShapes.
> As shown in Table 1, NVAE and VampPrior can achieve comparable reconstruction error compared to RecurD. However, their disentanglement qualities are not as good as RecurD, because RecurD additionally regularizes the inter-layer information sharing and alleviate the information redundancy of multiple layers. This experiment demonstrates the superiority of the proposed RecurD method in disentanglement compared to existing hierarchical VAEs. This study has been included in the Appendix D.1.
>
> |                | RecurD | $\beta$-VAE | NVAE   |VampPrior|
> | -------------- | ------ | ----------- | ------ | --------|
> | MIG            | 0.3105 | 0.2519      | 0.0081 | 0.1423 |
> | Reconst. Error |0.0083  | 0.0216      | 0.0078 | 0.0089 |
>
> *Table1: Comparison with hierarchical VAEs on 3DShapes.*
>
> > **Q2:** I would be a bit more careful with the tone on claims about the requirement of compositionality for disentanglement. Figure 1 is only an evidence in a toy example, not an actual demonstration. So statements as "fig 1 shows when ... is not effectively disentangled." (session 1) and "To achieve better disentanglement between ... their input feature sets .. are expected to be disentangled as demonstrated in our case study." (session 2.2) could be watered down a little.
>
> **A2:** Thanks for your comment. We have revised the two claims, also quoted below.
> Session 1: Figure 1\(b\) and Figure 1\(c\) show that the disentanglement quality of low-level features $m$ may impact the resulting representation $z$ in terms of disentanglement quality. This study demonstrates the potential benefit to regularize the compositional feature space of deep models during disentangled representation learning.
> Session 2.2: A disentangled representation of $m^l_i$ and $m^l_j$ may improve the disentanglement quality between $m^{l+1}_i$ and $m^{l+1}_j$.

---

> > ### Author Response · Authors · 2021-11-17
> > **Response to Reviewer yTV3 (Part 2/2)**
> >
> > > **Q3:** It's unclear if there are benefits from using MINE and the architecture in some of the experiments. For example, if I understood correctly, the beta-VAE objective yields better metrics on figure 6 than in figure 3.
> >
> > **A3:** Thanks for your comment. First we need to point out an incorrect description in Figure 6 caption. RecurD with $\lambda_2=0$ reduces to a variant of $\beta$-VAE with the compositional architecture, instead of the standard $\beta$-VAE. Therefore, the better results shown in Figure 6 indeed comes from using MINE and the compositional architecture. We have fixed Figure 6 caption in the manuscript accordingly.
> > To further validate the proposed method, we have conducted an extra ablation study. This study evaluates the following three variants of $\beta$-VAE ($\beta=8$) on 3DShapes, including **M1**: the vanilla $\beta$-VAE; **M2**: the vanilla $\beta$-VAE with MINE and the compositional architecture; **M3**: the proposed RecurD. As shown in Table 2, compared to the vanilla M1, M2 improves disentanglement quality via MINE and the compositional architecture. M3 achieves the best disentanglement quality by further introducing the proposed learning objective.
> >
> > |                |   M1   |   M2   |   M3   |
> > |:--------------:|:------:|:------:|:------:|
> > |      MIG       | 0.2565 | 0.2617 | 0.3105 |
> > | Reconst. Error | 0.0357 | 0.0389 | 0.0083 |
> >
> > *Table 2: Comparison of variant $\beta$-VAEs.*
> >
> > > **Q4:** The paper left me wondering what are the disentanglement metrics on the preceding layers, looking into MIG/SAP/DCI-D on $m^{L-1}$ and $m^{L-2}$ seems like a straightforward analysis that should be in the appendix or even in the main paper.
> >
> > **A4:** Thanks for your comment. Following your suggestion, we compute MIG on $m^{L-1}$ and $m^{L-2}$ of RecurD on 3DShapes. As shown in Table 3, high-level representations have higher MIG than that of low-level representations ($z > m^{L-1} > m^{L-2}$). These results confirm that the recursive propagation of inductive bias through the feed-forward network improves disentangled representation learning.
> >
> > |     | $m^{L-2}$ | $m^{L-1}$ | $z(m^{L})$ | Reconst. Error |
> > |:---:|:---------:|:---------:|:----------:|:--------------:|
> > | MIG |   0.1081  |   0.2892  |   0.3105   |     0.0083     |
> >
> > *Table 3: Comparison of MIG on $z$, $m^{L-1}$ and $m^{L-2}$.*
> >
> > > **Q5:** Some ablations on the architecture itself also seem to be missing (ablations on the loss are good). My intuition is that Gate of Mixture-of-Experts fits quite well with the disentanglement that we want, because of the top.
> >
> > **A5:** Thanks for your comment.  We incorporate the three suggested methods to evaluate the impact of architecture of Gate-of-Encoders (GoE), including **Linear**: GoE is implemented as a linear layer with softmax; **Fix**: GoE is implemented as fix assignment, i.e., we split $m_l$ in $d+1$ equal slices; and **Att**: GoE is implemented as a multi-head attention layer (the head is fixed as $8$). As shown in Table 4, this study demonstrates that GoE indeed fits well with the disentanglement process. This study is also included in the Appendix D.2.
> >
> > |                | Linear |   Fix  |   Att  | RecurD |
> > |:--------------:|:------:|:------:|:------:|:------:|
> > |       MIG      | 0.2539 | 0.2617 | 0.2604 | 0.3105 |
> > | Reconst. Error | 0.0176 | 0.0116 | 0.0095 | 0.0083 |
> > *Table 4: Ablation study on the GoE architecture.*
> >
> > ---
> > **Reference:**
> >
> > [1] [VAE with a VampPrior.](https://arxiv.org/abs/1705.07120) Tomczak, Jakub, and Max Welling. International Conference on Artificial Intelligence and Statistics. PMLR, 2018.
> >
> > [2] [Ladder variational autoencoders.](https://arxiv.org/abs/1602.02282) Sønderby, Casper Kaae, et al. Advances in neural information processing systems 29 (2016): 3738-3746.
> >
> > [3] [Nvae: A deep hierarchical variational autoencoder.](https://arxiv.org/abs/2007.03898) Vahdat, Arash, and Jan Kautz. arXiv preprint arXiv:2007.03898 (2020).

---

### Author Response · Authors · 2021-11-17
**General Response to Reviewers**

Thank you very much for reviewing our manuscript and providing detailed and constructive comments, which have been very helpful for us to improve the quality of our work. Enclosed, please see our answers to address the comments of individual reviewers.  In addition, we have uploaded a revised manuscript. Key changes in the manuscript are highlighted **in BLUE**. Unless otherwise specified, in this report, section, page, and line numbers correspond to those in the **revised manuscript**.

---

### Decision · Program_Chairs · 2022-01-20

**Decision:**

Accept (Poster)

**Comment:**

This paper proposes an algorithm for achieving disentangled representations by encouraging low mutual information between features at each layer, rather than only at the encoder output, and proposes a neural architecture for learning. Empirically, the proposed method achieves good disentanglement metric and likelihood (reconstruction error) in comparison to prior methods. The reviewers think that the methodology is natural and novel to their knowledge, and are happy with the detailed execution. The authors are encouraged to improve the presentation of the paper, by providing rigorous formulation of the "Markov chains" to avoid confusions, justification of the independence assumptions behind them, and more in-depth discussions of the learning objectives.